# Approaching Quartic Convergence Rates for Quasi-Stochastic Approximation with Application to Gradient-Free Optimization

**Caio Kalil Lauand**
University of Florida
caio.kalillauand@ufl.edu

**Sean Meyn**
University of Florida
meyn@ece.ufl.edu

## Abstract

Stochastic approximation is a foundation for many algorithms found in machine learning and optimization. It is in general slow to converge: the mean square error vanishes as $O(n^{-1})$. A deterministic counterpart known as quasi-stochastic approximation is a viable alternative in many applications, including gradient-free optimization and reinforcement learning. It was assumed in prior research that the optimal achievable convergence rate is $O(n^{-2})$. It is shown in this paper that through design it is possible to obtain far faster convergence, of order $O(n^{-4+\delta})$, with $\delta > 0$ arbitrary. Two techniques are introduced for the first time to achieve this rate of convergence. The theory is also specialized within the context of gradient-free optimization, and tested on standard benchmarks. The main results are based on a combination of novel application of results from number theory and techniques adapted from stochastic approximation theory.

## 1 Introduction

Stochastic approximation (SA) was introduced in the seminal work of Robbins and Monro [38]. The goal is to solve the root finding problem $\bar{f}(\theta^*) = 0$, in which $\bar{f} \colon \mathbb{R}^d \to \mathbb{R}^d$ is of the form

$$\bar{f}(\theta) := \mathsf{E}[f(\theta, \zeta)] \tag{1}$$

where $\zeta$ is a random vector taking values in $\mathbb{R}^m$. The basic algorithm is expressed as the $d$-dimensional recursion,

$$\theta_{n+1} = \theta_n + \alpha_{n+1} f(\theta_n, \zeta_{n+1}), \quad n \geq 0, \tag{2}$$

in which $\{\alpha_n\}$ is the step-size sequence, and $\zeta_{n+1} \xrightarrow{\mathrm{d}} \zeta$ as $n \to \infty$ (convergence in distribution). SA theory has attracted a great deal of attention over the past twenty years, motivated in large part by applications to reinforcement learning and optimization [49, 18, 32, 16, 7].

Convergence theory is couched in the *ODE Method* in which trajectories of (2) are compared to solutions of the ODE $\dot{\vartheta} = \bar{f}(\vartheta)$ (the *mean flow*). The major assumption required to ensure convergence to $\theta^*$, for each initial condition $\theta_0 \in \mathbb{R}^d$, is that the mean flow is globally asymptotically stable—see [7] for minimal assumptions on the "noise sequence" $\boldsymbol{\zeta}$. Establishing sharp rates of convergence is a far greater challenge. There is however a rich theory available to achieve the optimal rate of convergence for the mean-square error (MSE), which is in general $\mathsf{E}[\|\theta_n - \theta^*\|^2] = O(n^{-1})$.

There are many applications for which the designer of the algorithm also designs the noise. Notable examples include the introduction of exploration in reinforcement learning or gradient-free optimization. This motivates the use of *quasi-stochastic approximation* (QSA) in which the sequence $\boldsymbol{\zeta}$ is deterministic (e.g. mixtures of sinusoids or pseudo-random numbers). The idea was introduced in

36th Conference on Neural Information Processing Systems (NeurIPS 2022).

[20, 22], but has a much longer history in the context of gradient-free optimization—see [48, 26] for a survey of *extremum seeking control* (ESC).

Theory supporting rates of convergence of nonlinear QSA appeared only recently [9, 30]. Analysis and algorithms are posed in continuous time to simplify analysis. This setting is also motivated by recent success stories justifying algorithm design in continuous time, followed by a careful translation to obtain a discrete time algorithm. See for example theory surrounding acceleration methods of Polyak and Nesterov [2, 46, 19].

The notation adopted in [30, Chap. 4] will be used here: the *QSA ODE* is defined as

$$\tfrac{d}{dt}\Theta_t = a_t f(\Theta_t, \xi_t) \tag{3}$$

The deterministic continuous time process $\xi$ will be called the *probing signal*, and plays the role of $\zeta$ in SA; $a$ is called the gain process. The motivation for QSA is two-fold:

(i) It will be seen that the rate of convergence is far faster than SA, subject to careful choice of algorithm architecture.

(ii) In on-line applications the introduction of independent noise may not be realizable, or may impose unnecessary stress on equipment. In QSA design the components of the probing signal might be chosen to be sinusoidal signals of appropriate frequency and magnitude to ensure learning takes place without stress on physical devices.

*What is the optimal rate of convergence for QSA?* Consider the most basic one-dimensional problem in which $f(\Theta_t, \xi_t) = -\Theta_t + \xi_t$, with $\xi$ a zero-mean signal. The special case $a_t = 1/(1+t)$ results in an approximate average:

$$\Theta_T = \frac{1}{1+T}\Theta_0 + \frac{1}{1+T}\int_0^T \xi_t\, dt \tag{4}$$

If for example $\xi_t = \sin(\omega t)$ then the right hand side converges at rate $O(T^{-1})$, which translates to $O(T^{-2})$ for the "MSE"; this is far faster than the rate $O(T^{-1})$ expected for SA.

The special case of pure averaging is illustrated at the top in Fig. 1. The bold question mark in the figure refers to a question regarding a natural extension of the linear example in (4): *can we obtain the same rate of convergence for general non-linear QSA?* This question is posed and answered in the affirmative in [30, § 4.9], achieved through the averaging technique of Polyak and Ruppert, but only under conditions on the QSA ODE that could not be verified a priori. The current paper not only provides ways to ensure such conditions are always met, but also answers this question in a much more optimistic manner:

$$\sin(\omega t) \xrightarrow{\quad} \boxed{\text{AVG}} \xrightarrow{\frac{1}{\omega T}\left\{1 - \cos(\omega T)\right\}} = \theta^* + O(1/T)$$

$$\sin(\omega t) \xrightarrow{\quad} \boxed{\text{QSA}} \xrightarrow{\quad} = \theta^* + O(1/T) \quad \textbf{?}$$

Figure 1: What is the optimal convergence rate for QSA?

the presumption that MSE rates of order $O(T^{-2})$ are optimal for general QSA is *fallacy* since for any $\delta > 0$, rates of order $\|\Theta_T - \theta^*\|^2 = O(T^{-4+\delta})$ can be achieved through design.

The astonishingly fast MSE rates obtained for general QSA are a result of the *perturbative mean flow* (p-mean flow) representation for the QSA ODE. This representation expresses (3) in terms of the average vector field $\bar{f}$,

$$\tfrac{d}{dt}\Theta_t = a_t[\bar{f}(\Theta_t) - a_t\bar{\Upsilon}_t + \mathcal{W}_t], \qquad \mathcal{W}_t = \sum_{i=0}^{2} a_t^{2-i}\frac{d^i}{dt^i}\mathcal{W}_t^i \tag{5}$$

where $\{\mathcal{W}_t^i : i = 0, 1, 2\}$ are smooth functions of a larger state process, and $\bar{\Upsilon}_t = \bar{\Upsilon}(\Theta_t)$ with $\bar{\Upsilon}$ also smooth (see Thm. 2.1).

This representation is valuable only after boundedness of $\{\Theta_t\}$ is established, for which sufficient conditions are provided in [30, Prop. 4.33 and 4.34] (these conditions form part of Assumption (QSA3) in the supplementary material). A linearization of (5) around $\theta^*$ gives

$$\tfrac{d}{dt}\Theta_t = a_t[A^*(\Theta_t - \theta^*) - a_t\bar{\Upsilon}^* + \mathcal{W}_t + O(\|\Theta_t - \theta^*\|^2)] \tag{6}$$

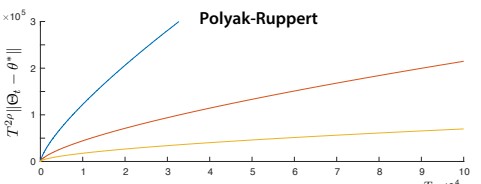
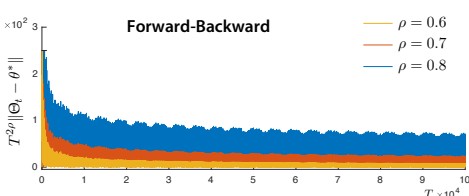

Figure 2: Rates of convergence for PR averaging and forward-backward algorithms.

where $A^* = \partial \bar{f}(\theta^*)$ and $\bar{\Upsilon}^* = \bar{\Upsilon}(\theta^*)$. The upper bound $\|\Theta_t - \theta^*\| = O(a_t)$ (previously obtained in [9, 30]) follows easily from (5).

Given estimates $\{\Theta_t : t \geq 0\}$, Polyak-Ruppert (PR) averaging defines the new estimates as follows,

$$\Theta_T^{\mathrm{PR}} = \frac{1}{T - T_0} \int_{T_0}^{T} \Theta_t \, dt\,, \quad T > T_0\,, \tag{7}$$

where the interval $[0, T_0]$ is known as the burn-in period. The time-average of $\{\mathcal{W}_t\}$ is of order $O(a_T^2)$, implying the rate $\|\Theta_T^{\mathrm{PR}} - \theta^*\|^2 = O(T^{-4+\delta})$ if and only if $\bar{\Upsilon}^* = 0$.

This brings us to the main contributions:

  (i) The p-mean flow representation (5) is introduced for the first time in Thm. 2.1.

 (ii) Convergence rates arbitrarily close to $O(T^{-2})$ can be obtained (so that the MSE is arbitrarily close to $O(T^{-4})$), which is far faster than the bound $O(T^{-1})$ assumed in prior research.

(iii) The near quartic rates for MSE are established for PR averaging in Thm. 2.2, subject to the assumption that $\bar{\Upsilon}^* = 0$. Sufficient conditions on the probing signal $\xi$ are provided in Thm. 2.1 to ensure that $\bar{\Upsilon}^* = 0$.

(iv) An alternative to PR averaging is introduced: *forward-backward filtering* achieves near quartic rates without the special conditions imposed in Thm. 2.2 (see Thm. 2.3).

The theory is refined in the context of gradient free-optimization, and the general theory is illustrated through numerical examples in this setting.

While theory is developed in continuous time, simulation studies using Euler approximations are consistent with theory. The plots shown in Fig. 2 are based on a two-dimensional example, whose details can be found in Section 2.1. The change of notation is also explained there: $(2\rho)^2 = 4 - \delta$, where $\rho$ is a design parameter subject to $\rho < 1$. The forward-backward algorithm is a new algorithm introduced in this paper, that achieves the $O(t^{-4+\delta})$ convergence rate; we find in these experiments that the same rates hold for the Euler approximation. Polyak-Ruppert averaging (7) cannot obtain this fast rate of convergence, because the frequencies used in this experiment violate a critical assumption in Thm. 2.2.

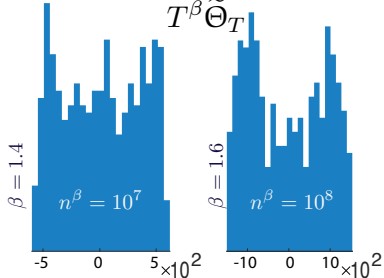

Figure 3: Quasi Monte Carlo using QSA with Polyak-Ruppert averaging. Histograms for $n = 10 \times T = 10^5$.

Fig. 3 illustrates how the techniques introduced in this paper specialize to quasi-Monte Carlo (QMC), based on two instances of the algorithm: one designed using $\beta = \sqrt{4 - \delta} = 1.4$, and the other with $\beta = 1.6$. Details are found in the supplementary material.

## 1.1 Preliminaries

A short literature survey and notation glossary are provided here.

**Literature review:** Polyak-Ruppert (PR) averaging was introduced in [39, 35, 36], and is now a standard workhorse in machine learning [14, 32, 31, 30].

Quasi-Monte Carlo (QMC) remains an active area of research for applications to estimation and optimization [33]. One aspect of the theory concerns techniques to construct a sequence $\{\zeta_i :$

$i \in \mathbb{Z}_+\} \subset [0,1]^K$ so that sample path averages are convergent, $n^{-1} \sum_{i=1}^n g(\zeta_i) \to \int g(x)\,\mu(dx)$ as $n \to \infty$ for functions $g\colon \mathbb{R}^K \to \mathbb{R}$, with $\mu$ uniform; theory predicts a convergence rate of $O(\log(n)^K/n)$ (see e.g. [1, Section 9.3]).

It is shown in Thm. B.1 of the present paper that the convergence rate $O(1/n)$ holds for QMC, subject to a smoothness condition on the function, and careful design of the probing signal. *The main results of this paper hinge on these new bounds for QMC, and further refinements.* These bounds are based in part on refinements of Baker's Theorem [28, 8].

QSA was first applied to finance in [20, 22]. Stability and convergence theory for QSA is recent [9, 30], following preliminary results in [3]; this work was motivated by applications to extremum seeking control, and successful application of QSA to Q-learning in [29].

The linear QSA ODE is treated in [40], where the first bounds on the rate of convergence of order $O(a_t)$ were obtained. This bound was extended to general nonlinear QSA in [9, 30], and it was also shown that the use of PR averaging results in a convergence rate of $O(T^{-1})$ subject to (up to now) unverifiable conditions.

QSA with fixed step-size is the topic of the contemporaneous work [24, 25]. An analog of the p-mean flow representation (5) is obtained, along with both steady-state and transient error bounds

Gradient-free optimization (GFO) concerns minimization of an objective function $\Gamma : \mathbb{R}^d \to \mathbb{R}$ based solely on measurements of $\Gamma(\theta)$ for selected values of $\theta \in \mathbb{R}^d$. The algorithms of Keifer and Wolfowitz are early examples [17, 6, 4, 15, 1]. The QSA ODEs for GFO considered in this paper are inspired by the simultaneous perturbation stochastic approximation (SPSA) algorithms of Spall [41, 42, 43]. Two examples are of the form (2), with

$$\text{1SPSA:} \qquad f^{\text{1SPSA}}(\theta_n, \zeta_{n+1}) = -\frac{1}{\varepsilon}\zeta_{n+1}\Gamma(\theta_n + \varepsilon\zeta_{n+1}) \tag{8a}$$

$$\text{2SPSA:} \qquad f^{\text{2SPSA}}(\theta_n, \zeta_{n+1}) = -\frac{1}{2\varepsilon}\big(\zeta_{n+1}[\Gamma(\theta_n + \varepsilon\zeta_{n+1}) - \Gamma(\theta_n - \varepsilon\zeta_{n+1})]\big) \tag{8b}$$

where $\varepsilon > 0$ and $\{\zeta_n\}$ is a zero-mean i.i.d. sequence.

These algorithms are biased in general unless the probing gain $\varepsilon$ is a vanishing function of $n$. There is substantial research in this setting: the best possible convergence rate for the mean square error is $O(n^{-\beta})$ with $\beta = (p-1)/(2p)$, provided the objective function is $p$-fold differentiable at $\theta^*$ [37]. Upper bounds appeared earlier in [13]. See [12, 11, 45, 34, 21] for more recent history.

Deterministic versions of SPSA were analyzed in [5] without sharp rates of convergence. The present paper follows the approach of [30, § 4.9], in which the applications to gradient-free optimization result in algorithms that resemble simple versions of the ESC algorithms surveyed in [48, 26] (see [25] for more on the relationship between ESC and QSA).

**Notation:** We restrict to probing signals that are nonlinear functions of sinusoids, of the form

$$\xi_t = G_0(\xi_t^0) \quad \text{with} \quad \xi_t^0 = [\cos(2\pi[\omega_1 t + \phi_1]), \cdots, \cos(2\pi[\omega_K t + \phi_K])]^\mathsf{T} \tag{9}$$

for which $G_0 : \mathbb{R}^K \to \mathbb{R}^m$ is analytic on $\mathbb{C}^K$, $\{\phi_i\}$ are arbitrary, and assumptions on the distinct frequencies $\{\omega_i\}$ will be imposed in the main results. When $G_0$ is linear we obtain the mixture of sinusoids

$$\xi_t = \sum_{i=1}^{K_\bullet} v^i \cos(2\pi[\omega_i t + \phi_i]), \quad \text{for vectors } \{v^i\} \subset \mathbb{R}^m \tag{10}$$

Denote $\Phi_t^i = \exp(2\pi j[\omega_i t + \phi_i])$ for each $i$ and $t$, which defines the $K$-dimensional vector-valued $\Phi_t$ function of time evolving on $\Omega := \{z \in \mathbb{C}^K : |z_i| = 1,\ 1 \le i \le K\}$. Writing $G(z) := G_0((z + z^{-1})/2)$ for non-zero $z \in \mathbb{C}^K$, we obtain $\xi_t = G(\Phi_t)$.

For two real-valued functions of time $\{\zeta_t^i : t \ge 0,\ i = 1, 2\}$ we denote

$$\langle \zeta^1, \zeta^2 \rangle := \lim_{T \to \infty} \frac{1}{T} \int_0^T \zeta_t^1 \zeta_t^2 \, dt, \qquad \langle \zeta^1 \rangle := \lim_{T \to \infty} \frac{1}{T} \int_0^T \zeta_t^1 \, dt, \qquad \tilde{\zeta}_t^i = \zeta_t^i - \langle \zeta^i \rangle \tag{11}$$

The notation is extended to vector or matrix valued functions. In particular, $\langle \xi \rangle = 0 \in \mathbb{R}^m$, and $\Sigma_\xi := \langle \xi \xi^\mathsf{T} \rangle$ is an $m \times m$ matrix.

The functions of time used in (11) are frequently obtained as functions of $(\Theta_t, \xi_t)$ [or more generally, $(\Theta_t, \Phi_t)$]. If $h \colon \mathbb{R}^d \times \mathbb{R}^m \to \mathbb{R}$ we often write $h_t \equiv h(\Theta_t, \xi_t)$ and $h_t^* \equiv h(\theta^*, \xi_t)$ to save space, and use the alternative notation $\bar{h} := \langle h \rangle$; identical notation is used for functions of $(\Theta_t, \Phi_t)$.

Let $\hat{h}$ denote a solution of *Poisson's equation* (12) with forcing function $h \colon \mathbb{R}^m \to \mathbb{R}$, which satisfies the defining equations,

$$\hat{h}(\Phi_T) = -\int_0^T \tilde{h}(\xi_t)\, dt + \hat{h}(\Phi_0)\,, \quad T \geq 0\,, \ \Phi_0 \in \Omega \tag{12}$$

It is assumed throughout that the solution is normalized so that its mean is zero. Thm. B.1 in the supplementary material provides conditions ensuring a smooth solution.

For a continuously differentiable ($C^1$) function $g \colon \mathbb{R}^d \times \Omega \to \mathbb{R}$, the directional derivative in the direction of the QSA vector field $f$ is denoted

$$[\mathcal{D}^f g](\theta, z) = \partial_\theta g\,(\theta, z) \cdot f(\theta, G(z))\,, \quad (\theta, z) \in \mathbb{R}^d \times \Omega \tag{13}$$

For a scalar-valued function $\delta_t > 0$, we use the notation $O(\delta_t)$ to denote a function of $t$; the notation indicates that there is a constant $B$ such that $\|O(\delta_t)\| \leq B\delta_t$ for all $t$. That is, we are only asserting an *upper bound*. For example, if $\delta_t = 1/(1+t)$ we are not claiming that $\|O(\delta_t)\| \leq B\delta_t^2$ is not possible.

**Organization:** Section 2 summarizes the main contributions of the paper, and Section 3 provides illustrations of the theory in application to GFO; in particular, convergence rate bounds are validated for GFO algorithms based on QSA. Conclusions and directions for future research are summarized in Section 4. Technical proofs and additional numerical results can be found in the supplemental material.

## 2 Towards Quartic Rates

### 2.1 Quasi-Stochastic Approximation

We begin with foundations regarding the QSA ODE (3) from [30, Ch. 4.5, 4.9]. The *mean vector field* is defined by the sample path average,

$$\bar{f}(\theta) := \lim_{T \to \infty} \frac{1}{T} \int_0^T f(\theta, \xi_t)\, dt\,, \quad \theta \in \mathbb{R}^d \tag{14}$$

Theory is based on comparison with the ODE,

$$\tfrac{d}{dt}\bar{\Theta}_t = a_t \bar{f}(\bar{\Theta}_t)\,, \quad \text{for } t \geq t_0 \text{ and } \bar{\Theta}_{t_0} = \Theta_{t_0}. \tag{15}$$

The time $t_0$ is fixed, but chosen suitably large in analysis of convergence rates.

The full list of assumptions (QSA0)–(QSA5) may be found in the supplementary material. We settle for a brief summary here: (QSA0) specifies conditions on the frequencies in (9). The assumptions allow the following special case: fix $\omega_1 > 0$ and an increasing sequence of positive integers $\{n_i\}$, and choose the probing signal of the form (9) with

$$\omega_i = n_i \omega_1 \ \text{ and } \ \phi_i = n_i \phi_1\,, \quad 2 \leq i \leq K_\bullet \tag{16}$$

See Prop. A.1 for explanation.

(QSA1) concerns the gain process; in the body of the paper we take $a_t = a_0(1+t)^{-\rho}$ for $\rho \in (1/2, 1]$ and a constant $a_0 > 0$. (QSA2) imposes global Lipschitz bounds on $f$ and $\bar{f}$. (QSA3) imposes global asymptotic stability of the mean flow $\tfrac{d}{dt}\vartheta_t = \bar{f}(\vartheta_t)$ and minor additional assumptions. (QSA4): $\bar{f}$ is $C^1$, $\bar{A}(\theta) = \partial_\theta \bar{f}\,(\theta)$ is bounded and Lipschitz continuous, and $A^* = A(\theta^*)$ is Hurwitz. (QSA5): existence of functions $\hat{f}$, and $\widehat{\Upsilon}$ that are Lipschitz continuous in $\theta$ solving for each $0 \leq t_0 \leq t_1$,

$$\hat{f}(\theta, \Phi_{t_0}) = \int_{t_0}^{t_1} \tilde{f}(\theta, \xi_t)\, dt + \hat{f}(\theta, \Phi_{t_1})\,, \qquad \tilde{f}(\theta, \xi_t) = f(\theta, \xi_t) - \bar{f}(\theta)$$

$$\hat{\hat{f}}(\theta, \Phi_{t_0}) = \int_{t_0}^{t_1} \hat{f}(\theta, \xi_t)\, dt + \hat{\hat{f}}(\theta, \Phi_{t_1}) \tag{17}$$

$$\widehat{\Upsilon}(\theta, \Phi_{t_0}) = \int_{t_0}^{t_1} \widetilde{\Upsilon}(\theta, \Phi_t)\, dt + \widehat{\Upsilon}(\theta, \Phi_{t_1})\,, \qquad \widetilde{\Upsilon}(\theta, \Phi_t) = \Upsilon(\theta, \Phi_t) - \bar{\Upsilon}(\theta)$$

where for $\widehat{A}(\theta, \xi) := \partial_\theta \hat{f}(\theta, \xi)$,

$$\Upsilon(\theta, \Phi_t) = -\widehat{A}(\theta, \Phi_t) f(\theta, \xi_t), \qquad \bar{\Upsilon}(\theta) = \lim_{T \to \infty} \frac{1}{T} \int_0^T \Upsilon(\theta, \Phi_t) \, dt \qquad (18)$$

If $A(\theta, \xi) = \bar{A}(\theta)$ for all $\theta$ and $\xi$ then $\bar{\Upsilon}(\theta^*) = 0$. This is a very restrictive special case, though it does hold for QMC for which $A(\theta, \xi) = -I$. In broad generality, we can always ensure $\bar{\Upsilon}(\theta^*) = 0$ by designing an algorithm with $\xi$ satisfying (QSA0) as shown in part (ii) of Thm. 2.1.

The solutions to Poisson's equation in (17) are used to define the terms in (5):

**Theorem 2.1.** *Suppose that (QSA1) holds and $a_t = (1+t)^{-\rho}$, with $\rho \in (0,1)$.*

(i) *Under (QSA5), the p-mean flow representation (5) holds with*

$$\mathcal{W}^0(\Theta_t, \Phi_t) = -[D^f \widehat{\Upsilon}](\Theta_t, \Phi_t) + \frac{r_t}{a_t}[D^f \hat{f}](\Theta_t, \Phi_t) \qquad (19a)$$

$$\mathcal{W}^1(\Theta_t, \Phi_t) = -[D^f \hat{f}](\Theta_t, \Phi_t) + \widehat{\Upsilon}(\Theta_t, \Phi_t) \qquad (19b)$$

$$\mathcal{W}^2(\Theta_t, \Phi_t) = \hat{f}(\Theta_t, \Phi_t), \qquad \text{where } r_t = \rho/(t+1) \qquad (19c)$$

(ii) *If (QSA0) and (QSA5) hold, then $\bar{\Upsilon}(\theta) = 0$ for each $\theta \in \mathbb{R}^d$.*

(iii) *If (QSA2) and (QSA3) hold, then $\{\Theta_t\}$ is ultimately bounded: there exists $b < \infty$ such that for any $\Theta_0$, $\limsup_{t \to \infty} \|\Theta_t\| \leq b$.*

*Proof:* The proof of part (i) is given at the end of Section B.3, while part (iii) follows from [30, Prop. 4.33 and 4.34].

Part (ii) is based on the geometry illustrated in Fig. 4: $S$ denotes the set of functions $g \colon \mathbb{R}^m \to \mathbb{R}$ that are analytic functions of the probing signal, and $\widehat{S}$ the set of functions $h \colon \Omega \to \mathbb{R}$ that have zero mean and solve Poisson's equation for some $g \in S$. Corollary B.6 establishes orthogonality of these two function classes in $L_2$: $\langle g, h \rangle = 0$ for each $g \in S$ and $h \in \widehat{S}$ (in the notation (11)), which indicates

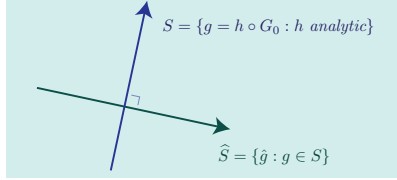

Figure 4: *Hidden geometry*: orthogonality of the function classes $S$ and $\widehat{S}$.

$$\lim_{T \to \infty} \int_0^T g(\xi_t) h(\Phi_t) \, dt = 0, \quad \text{for each } \Phi_0$$

The definition (18) gives

$$\Upsilon_i(\theta, \Phi) = -\sum_{j=1}^d \widehat{A}_{i,j}(\theta, \Phi) f_j(\theta, \xi), \quad \text{for each } \xi \in \mathbb{R}^m \text{ and each } \Phi \in \Omega.$$

Since $\widehat{A}_{i,j}(\theta, \cdot) \in \widehat{S}$ and $f_j(\theta, \cdot) \in S$ for each $i, j$ and $\theta$, it follows that $\bar{\Upsilon}(\theta) = 0$. $\qquad \square$

## 2.2 Acceleration

Two approaches to obtain the convergence rate bounds $\|\Theta_T - \theta^*\| = O(a_T^{2\rho})$ are described here.

**Polyak-Ruppert averaging** Fast convergence can be obtained using standard averaging, but only under restrictive conditions. In Thm. 2.2 we adopt the notation,

$$\bar{Y}^* := [A^*]^{-1} \bar{\Upsilon}^*, \quad \text{where } \bar{\Upsilon}^* = \bar{\Upsilon}(\theta^*), \qquad \hat{f}_t^* := \hat{f}(\theta^*, \Phi_t), \ t \geq 0 \qquad (20)$$

The proof of Thm. 2.2 is postponed to the supplementary material.

**Theorem 2.2.** *Suppose that (QSA1)–(QSA5) hold, and the following additional assumptions are imposed: $\{\Theta_t\}$ is obtained using $a_t = (1+t)^{-\rho}$ with $\rho \in (1/2, 1)$, and estimates using PR averaging (7) are obtained using $T_0 = (1 - 1/\kappa)T$ with $\kappa > 1$ fixed. Then,*

$$\Theta_t = \theta^* + a_t[\bar{Y}^* - \hat{f}_t^*] + o(a_t) \qquad (21a)$$

$$\Theta_T^{\mathsf{PR}} = \theta^* + a_T[c(\kappa, \rho) + o(1)]\bar{Y}^* + O(T^{-2\rho}) \qquad (21b)$$

$$\text{with } c(\kappa, \rho) = \kappa[1 - (1 - 1/\kappa)^{1-\rho}]/(1 - \rho)$$

Thm. 2.1 and the definition (20) imply that $\bar{Y}^* = 0$ under (QSA0). Eq. (21b) of Thm. 2.2 then implies that PR averaging leads to a convergence rate bounded by $O(T^{-2\rho})$, subject to (QSA0)–(QSA5).

It was discovered recently that the vector $\bar{\Upsilon}^*$ also appears in representations of the SA recursion (2) when $\{\zeta_n\}$ is a smooth function of a Markov chain. This introduces bias and potentially large variance for fixed step-size algorithms ($\alpha_{n+1} \equiv \alpha > 0$), even when using averaging [24, §2.5.3],[23].

**Forward-backward filtering**  We present a technique to achieve near quartic MSE convergence rate without imposing additional structure on the probing signal. Motivation is two-fold: first, it is hoped that this technique can be extended to SA for which there is no known analog to (QSA0); second, in some applications of QSA it may not be possible to ensure that (QSA0) is satisfied.

Forward-backward (FB) filtering is defined by a pair of QSA ODEs:

$$\tfrac{d}{dt}\Theta_t^- = a_t f(\Theta_t^-, \xi_t^-) \qquad \Theta_T^{\mathsf{FB}} = \tfrac{1}{2}[\Theta_T^{\mathsf{PR}} + \Theta_T^{\mathsf{PR}^-}] \tag{22}$$

where $\xi_t^- := \xi_{-t}$, $\Theta_T^{\mathsf{PR}}$ is defined in (7), and $\Theta_T^{\mathsf{PR}^-}$ is defined analogously using $\{\Theta_t^-\}$.

**Theorem 2.3.** *If the assumptions of Thm. 2.2 hold, then, $\Theta_T^{\mathsf{FB}} = \theta^* + O(T^{-2\rho})$.*

*Overview of proof:* The analogous vector $\bar{Y}^*$ for the QSA ODE (22) is denoted $\bar{Y}^{*-}$. The identity $\bar{Y}^{*-} = -\bar{Y}^*$ is established through consideration of (20), and the desired bound is then obtained by combining (21b) with the definition (22). $\qquad\square$

Assumption (QSA0) implies that the frequencies $\{\omega_i\}$ are irrationally related. This might appear far stronger than needed: for one, the full rank condition for $\Sigma_\xi$ can be achieved under far weaker assumptions. In two dimensions, $\Sigma_\xi = \frac{1}{2}I$ for each of the following special cases: $\xi_t = [\cos(t), \cos(2t)]^\mathsf{T}$ and $\xi_t = [\cos(t), \sin(t)]^\mathsf{T}$. However, in each case we can construct a second order polynomial $h: \mathbb{R}^2 \to \mathbb{R}$ such that $h(\xi_t) = 0$ for all $t$, so that the *excitation* implied by the full rank condition is lost through a simple nonlinearity. This is only the first sign of trouble.

We consider next a numerical example designed with four positive frequencies that are linearly independent over the rationals, denoted $[\omega_1, \omega_2, \omega_3, \omega_4]$. Consider the linear QSA ODE,

$$\tfrac{d}{dt}\Theta_t = a_t f(\Theta_t, \xi_t) = (1+t)^{-\rho}[(A^* + A_t^\circ)\Theta_t + 10b_t^\circ]$$

$$\text{with} \quad A_t^\circ = \begin{bmatrix} 4\sin(\omega_1 t) & \sin(\omega_2 t) \\ \sin(\omega_3 t) & 4\sin(\omega_4 t) \end{bmatrix} \qquad b_t^\circ = \begin{bmatrix} 2\cos(\omega_1 t) \\ \cos(\omega_4 t) \end{bmatrix} \qquad \text{and } A^* \text{ Hurwitz.}$$

We have $\bar{f}(\theta) = A^*\theta$ and hence $\theta^* = 0$. To fit the notation (9) we must take $\xi_t \in \mathbb{R}^6$, which can be chosen so that $\Sigma_\xi = \frac{1}{2}I$. The expression $\bar{Y}^* = -20[2/\omega_1, 1/\omega_4]^\mathsf{T}$ follows from (20).

The conclusions of Thm. 2.3 are illustrated in Fig. 2. The model was simulated with $A^* = -0.8I$ and frequencies $[\pi, \sqrt{3}, 4, \sqrt{5}]/5$. The QSA ODEs were constructed using three values of the parameter in the gain process, $\rho \in \{0.6, 0.7, 0.8\}$.

The plots shown on the left in Fig. 2 were obtained using PR averaging with $T_0 = (1 - 1/\kappa)T$ and $\kappa = 4$. The convergence rate is $O(T^{-\rho})$ because $\bar{Y}^* \neq 0$ (recall (21b)). FB filtering achieves the convergence rate of $O(T^{-2\rho})$, for each value of $\rho$ tested, as predicted by Thm. 2.2.

## 3  Gradient-Free Optimization

We now turn to applications to gradient-free optimization (GFO). It is assumed that the objective $\Gamma: \mathbb{R}^d \to \mathbb{R}$ is $C^2$ and that it has a unique minimizer denoted $\theta^{\mathsf{opt}}$, and that $\Sigma_\xi := \langle \xi\xi^\mathsf{T} \rangle$ is full rank.

### 3.1  Gradient-Free Optimization and QSA

Two approaches are considered in the following: For $\varepsilon > 0$ and each $\Theta_0 \in \mathbb{R}^d$,

$$\textbf{1qSGD:} \qquad f^{1\mathsf{Q}}(\Theta_t, \xi_t) = -\frac{1}{\varepsilon}\xi_t\Gamma(\Theta_t + \varepsilon\xi_t) \tag{23}$$

$$\textbf{2qSGD:} \qquad f^{2\mathsf{Q}}(\Theta_t, \xi_t) = -\frac{1}{2\varepsilon}\xi_t[\Gamma(\Theta_t + \varepsilon\xi_t) - \Gamma(\Theta_t - \varepsilon\xi_t)] \tag{24}$$

The mean vector fields $\bar{f}^{1Q}, \bar{f}^{2Q}$ are identical, provided the probing signal is symmetric:

**Proposition 3.1.** *Suppose (QSA1)-(QSA5) hold, that $\Gamma$ is $C^2$ with unique minimizer $\theta^{\mathrm{opt}}$. Assume moreover that the probing signal is of the form (10) with $m = d$ and distinct frequencies $\{\omega_i\}$. Then, the average vector fields for 1qSGD and 2qSGD are equal:*

$$\bar{f}(\theta) = \langle f^{1Q}(\theta, \xi) \rangle = \langle f^{2Q}(\theta, \xi) \rangle = -\Sigma_\xi \nabla \Gamma(\theta) + O(\varepsilon^2) \tag{25}$$

For the special case of a strongly convex objective $\Gamma$, bounds on bias of 2qSGD are well known [44]. An application of Prop. 3.1 implies that the bias for 1qSGD is identical:

**Corollary 3.2.** *Suppose the assumptions of Prop. 3.1 hold, and $\Gamma$ is strongly convex. Then, $\|\theta^* - \theta^{\mathrm{opt}}\| \le O(\varepsilon^2)$ for either 1qSGD or 2qSGD.*

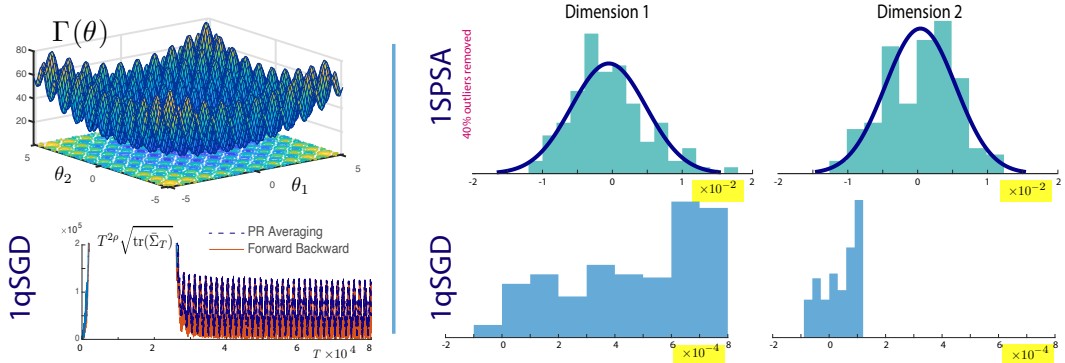

Figure 5: Rastrigin objective (top left), scaled trace of empirical covariance (bottom left), histograms of estimation error for 1SPSA with PR averaging (top middle and top right), histograms of estimation error for 1qSGD with PR averaging (bottom middle and bottom right).

## 3.2 Numerical Examples

The numerical results surveyed here and in the supplementary material used a crude approximation of the QSA ODE: an Euler scheme with sampling interval of 1 sec. It is found that the results remain consistent with theory for moderate dimension. A metric for testing this theory was via the empirical covariance, obtained using $M \ge 50$ independent runs to obtain $\Theta_T^i : \{1 \le i \le M\}$, and then

$$\overline{\Sigma}_T = \frac{1}{M} \sum_{i=1}^M \Theta_T^i \Theta_T^{i\,\mathsf{T}} - \bar{\Theta}_T \bar{\Theta}_T^{\mathsf{T}}, \qquad \bar{\Theta}_T = \frac{1}{M} \sum_{i=1}^M \Theta_T^i \tag{26}$$

Thm. 2.2 and Thm. 2.3 predict that the root mean square error $T^{2\rho}\sqrt{\mathrm{tr}(\overline{\Sigma}_T)}$ is bounded in $T$, subject to conditions.

**Rastrigin:** The algorithms were tested in this standard benchmark [47], for which a plot of the objective can be found on the upper left in Fig. 5. In each qSGD ODE the values $\varepsilon = 0.25$ and $\rho = 0.85$ were used, with $\kappa = 5$ in PR averaging. In the SPSA stochastic counterparts the exploration sequence $\{\zeta_n\}$ was chosen i.i.d. and zero-mean, with values in $\{\pm\sqrt{2}\}$ (these values were chosen so that the covariance matrices for both SPSA and qSGD would equal). Further details on the experiment are postponed to the supplementary material.

As predicted by theory, the root-MSE $T^{2\rho}\mathrm{tr}(\overline{\Sigma}_T)^{\frac{1}{2}}$ is bounded in $T$ when using FB filtering or PR averaging: see the plot on the lower left in Fig. 5. Histograms are shown only for 1qSGD and 1SPSA using PR averaging. The results

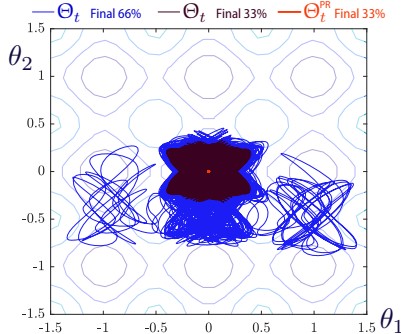

Figure 6: $\{\Theta_T\}$ for the final 66% of the run, and $\{\Theta_T^{\mathrm{PR}}\}$ for the final 33%.

obtained for 1qSGD with FB filtering were similar. Results obtained using 2qSGD and 2SPSA were similar qualitatively. Outliers, identified with Matlab's *isoutlier* function, are excluded in these plots. Outliers were found in 20% of the independent runs for 1SPSA, and none for 1qSGD. Outliers for deterministic algorithms were observed in other experiments, but fewer than in their stochastic counterparts.

Fig. 6 shows part of the trajectory of $\{\Theta_T\}$ for a short run. We see that $\{\Theta_T\}$ oscillates between saddle points and local extrema before settling around the minimizer $\theta^{\mathsf{opt}} = 0$ near the end of the run. The corresponding PR estimates very closely approximate $\theta^{\mathsf{opt}}$ for the final 33% of the run.

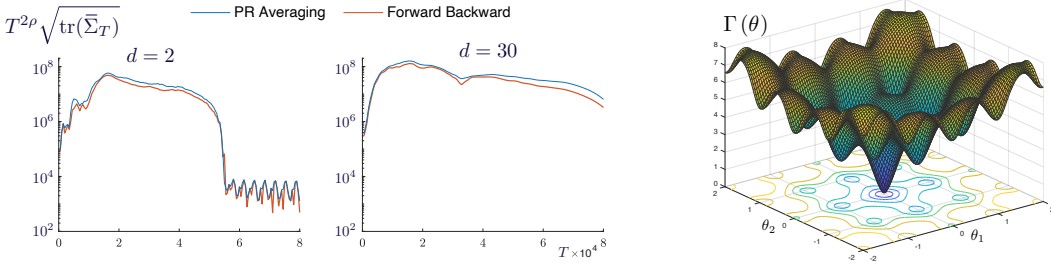

Figure 7: Ackley objective for $d = 2$ (right). Scaled root-MSE for $d = 2$ (left) and $d = 30$ (middle).

**Ackley:** This is another standard benchmark [47]. Its objective function is shown on the right in Fig. 7 in the special case $d = 2$.

Fig. 7 shows the scaled root-MSE for dimensions $d = 2$ and $d = 30$ with $\rho = 0.85$. Both PR averaging and FB filtering are successful in achieving $O(T^{-2\rho})$ convergence rates. We observe much larger values of $\mathrm{tr}\,(\bar{\Sigma}_T)^{\frac{1}{2}}$ at the end of the run for $d = 30$. It is possible that better results will be improved by with a different choice of probing signal $\xi$.

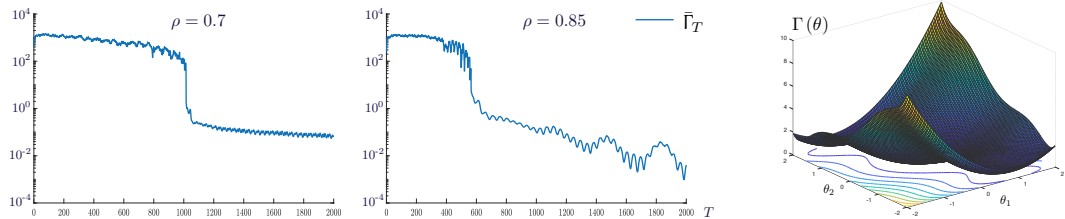

Figure 8: Average of $\Gamma(\Theta_T^i)$ over $1 \le i \le M$ independent runs for the three-hump camel.

**Three-Hump Camel:** This is another standard benchmark [47], whose objective function is plotted on the right in Fig. 8. Results from experiments using PR averaging, surveyed in Section C.2, show that the root-MSE is similar to what is obtained for the Ackley objective with $d = 2$. In this example the value $\Gamma(\Theta_t^{\mathsf{PR}})$ converges rapidly towards its optimal value of zero.

## 4 Conclusions

While it is convenient to design exploration around i.i.d. signals, and this approach opens the doors to many powerful tools from probability theory, we have shown that deterministic "noise" has significant benefits. Convergence rates can be accelerated dramatically provided the algorithm and deterministic probing signals are chosen with care.

There are many avenues for future research:

$\triangle$ The impact of dimension on convergence rate appears to be understood for QMC [1]. We currently do not know how to extend this theory to QSA because our analysis is rooted in properties of the $K$-dimensional clock process $\Phi$, which is far removed from the QMC setting.

$\triangle$ The optimal convergence rate for QSA is unknown for the approaches described in this paper, and we currently lack universal bounds that are not restricted to a particular algorithm.

(i) Can we improve the upper bounds for PR averaging and the FB algorithm? It may be possible to achieve the bound $O(T^{-1-\rho})$, rather than $O(T^{-2\rho})$ in current theory.

(ii) Can we find fundamental bounds for any algorithm? *We haven't ruled out the creation of an algorithm with rate of convergence of order $O(T^{-10})$.* This question is motivated by a long history of success in the stochastic approximation literature [10, 39, 35, 36, 37].

$\triangle$  It is not clear that the constraint $\rho > \frac{1}{2}$ is required in this deterministic setting, and may be removed if we can improve the bounds in our analysis. We have found in some experiments that the use of PR averaging results in very fast convergence even when this constraint is violated.

$\triangle$  Under what conditions is $\sqrt{\text{tr}(\overline{\Sigma}_T)}$ strictly smaller using the FB algorithm as compared to PR averaging? We find in experiments that the FB algorithm usually outperforms averaging, but we have yet to find tools to obtain bounds that are rich enough to compare the two approaches.

**Acknowledgements**  Many thanks to Profs. Yann Bugeaud (U. Strasbourg) and Gisbert Wüstholz (ETH) for guidance on the literature surrounding Baker's Theorem.

Financial support from ARO award W911NF2010055 and National Science Foundation award EPCN 1935389 is gratefully acknowledged

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
