# Supplementary Material

## Contents

## A   Assumptions for QSA Theory

The assumptions imposed in [30, §4.5, 4.9] are listed below. The setting there is far more general than here, since the entries of the probing signal are not restricted to functions of sinusoids. In this prior work it is assumed that the probing signal is a function of a deterministic signal $\mathbf{\Phi}$ of the form $\xi_t = G(\Phi_t)$, where $\mathbf{\Phi}$ is the state process for a dynamical system,

$$\tfrac{d}{dt}\Phi_t = \mathrm{H}(\Phi_t) \tag{27}$$

with $\mathrm{H}\colon \Omega \to \Omega$ and $\Omega$ is a compact subset of the Euclidean space. It is assumed that it has a unique invariant measure $\pi$ on $\Omega$.

For the special case treated here, with $\Phi_t^i = \exp(2\pi j[\omega_i t + \phi_i])$ for each $i$ and $t$, we have $\Omega = S^K \subset \mathbb{C}^K$ with $S$ the unit circle in $\mathbb{C}$. The dynamics (27) and the function $G$ are:

$$\begin{aligned}
\tfrac{d}{dt}\Phi_t &= W\Phi_t & W &:= 2\pi j\mathrm{diag}(\omega_i) \\
\xi_t &= G(\Phi_t) & G(z) &:= G_0(\tfrac{1}{2}(z + z^{-1})), \qquad z \in \mathbb{C}^K
\end{aligned} \tag{28}$$

where $G_0$ is the function appearing in (9). The function $G$ is then analytic on $z \in \{\mathbb{C} \setminus \{0\}\}^K$ if $G_0$ is analytic on $\mathbb{C}^K$.

The proof of Prop. B.2 below begins with a proof that $\pi$ exists, with density $\rho(z_1, \ldots, z_K) = \prod_{i=1}^{K} u(z_i)$, where $u$ denotes the uniform distribution on $S$. In particular, since $\xi_t = G(\Phi_t)$ for some function $G$, the function $\bar{f}$ in (14) can be expressed,

$$\bar{f}(\theta) = \int_{\Omega} f(\theta, G(z)) \, \pi(dz) \tag{29}$$

For any function $h : \mathbb{R}^d \times \Omega \to \mathbb{R}$ that is $C^1$, we define

$$\mathcal{D}h \, (\theta, z) := a_t [\mathcal{D}^f h](\theta, z) + \partial_z h \, (\theta, z) \cdot W z \,, \quad (\theta, z) \in \mathbb{R}^d \times \Omega \tag{30}$$

where $\mathcal{D}^f h$ is defined in (13). The function $\mathcal{D}h$ is continuous and the functional $\mathcal{D}$ is known as the *differential generator* in the Markov literature. Upon denoting $g = \mathcal{D}h$, the chain rule gives

$$g(\Theta_t, \Phi_t) = \tfrac{d}{dt} h(\Theta_t, \Phi_t) \tag{31}$$

The remaining assumptions are listed below. Lipschitz bounds on $\hat{f}$ and those in (QSA5) are partially justified by Thm. B.1 subject to smoothness assumptions on $f$.

The first assumption sets restrictions on frequencies.

**(QSA0a)** $\xi_t = G_0(\xi_t^0)$ for all $t$, with $\xi_t^0$ defined in (9). The function $G_0 : \mathbb{R}^K \to \mathbb{R}^m$ is assumed to be analytic, with the coefficients in the Taylor series expansion for $G_0(\xi_t^0)$ absolutely summable.

**(QSA0b)** The frequencies $\{\omega_1, \ldots, \omega_K\}$ are chosen of the form

$$\begin{aligned} \omega_i &= \log(a_i/b_i) > 0 \,, \ 1 \le i \le K \,, \\ \{\omega_i\} &\text{ , linearly independent over the rationals,} \end{aligned} \tag{32a}$$

and with $\{a_i, b_i\}$ positive integers.

**(QSA1)** The process $\boldsymbol{a}$ is non-negative, monotonically decreasing, and

$$\lim_{t \to \infty} a_t = 0, \qquad \int_0^{\infty} a_r \, dr = \infty. \tag{32b}$$

**(QSA2)** The functions $\bar{f}$ and $f$ are Lipschitz continuous: for a constant $L_f < \infty$,

$$\|\bar{f}(\theta') - \bar{f}(\theta)\| \le L_f \|\theta' - \theta\|,$$

$$\|f(\theta', \xi) - f(\theta, \xi)\| + \|f(\theta, \xi') - f(\theta, \xi)\| \le L_f [\|\theta' - \theta\| + \|\xi' - \xi\|] \,, \quad \theta', \theta \in \mathbb{R}^d \,, \xi, \xi' \in \mathbb{R}^m$$

**(QSA3)** The ODE $\tfrac{d}{dt} \vartheta_t = \bar{f}(\vartheta_t)$ is globally asymptotically stable with unique equilibrium $\theta^*$. Moreover, one of the following conditions holds:

(a) There is a Lipschitz continuous Lyapunov function $V : \mathbb{R}^d \to \mathbb{R}_+$, a constant $\delta_0 > 0$ and a compact set $\mathsf{S}$ such that $\nabla V(\vartheta_t) \cdot \bar{f}(\vartheta_t) \le -\delta_0 \|\vartheta_t\|$ whenever $\vartheta_t \notin \mathsf{S}$.

(b) The scaled vector field $\bar{f}_\infty : \mathbb{R}^d \to \mathbb{R}^d$ defined by $\bar{f}_\infty(\theta) := \lim_{c \to \infty} \bar{f}(c\theta)/c, \theta \in \mathbb{R}^d$, exists as a continuous function. Moreover, the ODE@$\infty$ defined by

$$\tfrac{d}{dt} x_t = \bar{f}_\infty(x_t) \tag{32c}$$

is globally asymptotically stable [30, §4.8.4].

**(QSA4)** The vector field $\bar{f}$ is differentiable, with derivative denoted

$$\bar{A}(\theta) = \partial_\theta \bar{f}(\theta) \tag{32d}$$

That is, $\bar{A}(\theta)$ is a $d \times d$ matrix for each $\theta \in \mathbb{R}^d$, with $\bar{A}_{i,j}(\theta) = \dfrac{\partial}{\partial \theta_j} \bar{f}_i(\theta)$.

Moreover, the derivative $\bar{A}$ is Lipschitz continuous, and $\bar{A}^* = \bar{A}(\theta^*)$ is Hurwitz.

**(QSA5)** $\Phi$ satisfies the following ergodic theorems for the functions of interest, for each initial condition $\Phi_0 \in \Omega$:

(i) For each $\theta$ there exists a solution $\hat{f}(\theta, \cdot)$ to *Poisson's equation* with forcing function $f$. That is,

$$\hat{f}(\theta, \Phi_{t_0}) = \int_{t_0}^{t_1} [f(\theta, \xi_t) - \bar{f}(\theta)]\, dt + \hat{f}(\theta, \Phi_{t_1}), \quad 0 \le t_0 \le t_1 \tag{32e}$$

with $\bar{f}$ given in (29) and for each $\theta$, $\int_\Omega \hat{f}(\theta, z)\, \pi(dz) = 0$. Finally, $\hat{f}$ is continuously differentiable $(C^1)$ on $\mathbb{R}^d \times \Omega$. Its Jacobian with respect to $\theta$ is denoted

$$\widehat{A}(\theta, z) := \partial_\theta \hat{f}(\theta, z) \tag{32f}$$

$$\text{where } \int_\Omega \widehat{A}(\theta, z)\, \pi(dz) = 0 \quad \text{for each } \theta \in \mathbb{R}^d \tag{32g}$$

(ii) For each $\theta$, there are $C^1$ solutions to Poisson's equation with forcing functions $\hat{f}$ and $\Upsilon$. They are denoted $\hat{\hat{f}}$ and $\widehat{\Upsilon}$, respectively, satisfying

$$\hat{\hat{f}}(\theta, \Phi_{t_0}) = \int_{t_0}^{t_1} \hat{f}(\theta, \xi_t)\, dt + \hat{\hat{f}}(\theta, \Phi_{t_1}) \tag{32h}$$

$$\widehat{\Upsilon}(\theta, \Phi_{t_0}) = \int_{t_0}^{t_1} [\Upsilon(\theta, \Phi_t) - \bar{\Upsilon}(\theta)]\, dt + \widehat{\Upsilon}(\theta, \Phi_{t_1}), \quad 0 \le t_0 \le t_1 \tag{32i}$$

$$\text{with} \quad \bar{\Upsilon}(\theta) = \langle \Upsilon(\theta, \Phi) \rangle = - \int_\Omega \widehat{A}(\theta, z) f(\theta, G(z))\, \pi(dz) \tag{32j}$$

Moreover, for each $\theta$,

$$\int_\Omega \hat{\hat{f}}(\theta, z)\, \pi(dz) = \int_\Omega \widehat{\Upsilon}(\theta, z)\, \pi(dz) = 0$$

**Proposition A.1.** *For any integer $n \ge 2$ there is a polynomial $\varrho$ satisfying*

$$\varrho(\cos(r)) = \cos(nr), \quad r \ge 0$$

*Consequently, in the special case of (QSA0b) with $K = 1$, for any increasing sequence of positive integers $n_1 = 1 < n_2 < \cdots < n_{K_\bullet}$ there is a polynomial function $G_0 \colon \mathbb{R}^K \to \mathbb{R}^m$ such that*

$$\xi_t = G_0(\omega_1 t + \phi_1) = \sum_{i=1}^{K_\bullet} v^i \cos(2\pi[n_i \omega_1 t + n_i \phi_1])$$

*with $\{v^i\}$ as in (10).*

Prop. A.1 tells us that if we choose a small value for $K$ in (QSA0) then we must impose stronger conditions on the values of phases for the mixture of sinusoids model (10).

## B Technical Proofs

### B.1 Tighter Bounds for Quasi-Monte Carlo

Thm. B.1 below justifies boundedness of the integral (12) and much more. Through careful design of the frequencies $\{\omega_i\}$ appearing in (9) we can apply refinements of Baker's Theorem, as surveyed in the monograph [8].

**Theorem B.1.** *Suppose that the function $h \colon \mathbb{R}^m \to \mathbb{R}$ is analytic in a neighborhood of the unit hypercube $[-1, 1]^m$, and that $\xi_t$ in (9) satisfies (QSA0). Then the limit defining $\langle h \rangle$ in (11) exists with $h_t = h(\xi_t)$, and the following ergodic bounds hold for some constant $B_f$ independent of the phase values $\{\phi_i\}$:*

$$\left| \frac{1}{T} \int_0^T \tilde{h}(\xi_t)\, dt \right| \le B_f \frac{1}{T}, \quad T > 0 \quad \text{where } \tilde{h} = h - \langle h \rangle. \tag{33}$$

Moreover, there is a function $\hat{h} \colon \mathbb{C}^K \to \mathbb{C}^K$ that is analytic in a neighborhood of the domain $\Omega := \{z \in \mathbb{C}^K : |z_i| = 1,\, 1 \le i \le K\}$, real-valued on $\Omega$, and satisfying

$$\int_0^T \tilde{h}(\xi_t)\, dt = \hat{h}(\Phi_0) - \hat{h}(\Phi_T)\,, \quad T \ge 0\,,$$

$$\langle \hat{h}(\Phi) \rangle = \lim_{T \to \infty} \frac{1}{T} \int_0^T \hat{h}(\Phi_t)\, dt = 0 \tag{34}$$

A version of (33) in discrete time is contained in Thm. B.10. This result is a big surprise, given that there is so much theory predicting $O(\log(T)^K/T)$ bounds. Much more surprising is that $O(1/T)$ is a very poor rate of convergence for the algorithms developed in this paper, as observed in Fig. 3. This is explained in Section C.1, along with examples to show how the theory can be applied to obtain convergence rates of order $O(T^{-2+\delta})$ for specially designed QSA algorithms, and in particular new approaches to QMC.

A few key ideas are presented here: The assumption that $h$ is analytic is imposed so that we can first restrict to complex exponentials, $H(\Phi_t) = \exp(2\pi j t \omega t)$, whose integral equals $(2\pi j\omega)^{-1} H(\Phi_t)$ when $\omega \ne 0$. In the proof of Thm. B.1 we use $\omega = \sum_{i=1}^K n_i \omega_i$ with $\{n_i\}$ integers; they are not necessarily positive, but at least one $n_i$ is assumed non-zero. Extensions of Baker's Theorem, as surveyed in [8], give a strict lower bound of the form $|\omega| \ge \delta n^{-C}$, where $n = 3 + \sum |n_i|$ and $\delta, C$ are non-negative constants that are independent of $n$—see Thm. B.7. This combined with routine Taylor series bounds establishes the desired conclusions.

Bounds on the constant $B_f$ requires bounds on the constant $C$ appearing in Thm. B.7. Current bounds on this constant grow rapidly with $K$, such as the doubly exponential bounds obtained in [27] and [28]. Recall from Prop. A.1 that the nonlinearity $G_0$ in (28) permits the creation of rich probing signals from simple ones. Hence, *large dimension $d$ does not mean that we require a large value of $K$.*

**An Elementary Ergodic Theorem** We begin with an alternative characterization of $\overline{h} = \langle h \rangle$.

The ergodic theorems presented here are based upon a stationary relaxation of the solution to the ODE $\frac{d}{dt} \Phi_t = W \Phi_t$ (recall (28)). We do not require (QSA0) here, but we do require that the frequencies $\{\omega_i : 1 \le i \le K\}$ are *distinct*.

Suppose that the initial conditions $\{\Phi_0^i : 1 \le i \le K\}$ are chosen *randomly* with i.i.d. values uniform on the unit circle $S \subset \mathbb{C}$. It follows that $\{\Phi_t^i : 1 \le i \le K\}$ remain i.i.d. with uniform distribution for each $t \in \mathbb{R}$, so that $\mathbf{\Phi} = \{\Phi_t : -\infty < t < \infty\}$ is a stationary Markov process. Stationarity implies the Law of Large Numbers: for any Borel measurable function $H \colon S^K \to \mathbb{C}$,

$$\lim_{T \to \infty} \frac{1}{T} \int_0^T H(\Phi_t)\, dt = \lim_{T \to \infty} \frac{1}{T} \int_0^T H(\Phi_{-t})\, dt = \mathrm{E}[H(\Phi_0)] = \int_\Omega H(z)\, dz \qquad a.s. \tag{35}$$

Conditioning on $\Phi_0 = z^0$ we can extend this limit to a.e. initial condition $z^0 \in \Omega := S^K$.

For this stationary realization of $\mathbf{\Phi}$, we henceforth regard $\{\xi_t = G(\Phi_t) : t \in \mathbb{R}\}$ as a *steady-state* realization of the probing signal.

Consider any real-valued function $h$ of the probing signal (9). We have $h(\xi_t) = H(\Phi_t)$ with $H := h \circ G$ from (28), which then implies a law of large numbers. A characterization of the limit is obtained in the following, along with a relaxation of the smoothness assumption imposed in Thm. B.1.

**Proposition B.2.** *Suppose that the probing signal is defined using (9), with distinct frequencies $\{\omega_i : 1 \le i \le K\}$. Consider any Borel measurable function $h \colon \mathbb{R}^m \to \mathbb{R}$ satisfying $\mathrm{E}[|h(G_0(X))|] < \infty$, where the $K$-dimensional random vector has independent entries, with common distribution equal to the arcsine law on $[-\pi, \pi]$.*

*The following limits then hold for a.e. set of phase angles $\{\phi_i\}$:*

$$\lim_{T \to \infty} \frac{1}{T} \int_0^T h(\xi_t)\, dt = \lim_{T \to \infty} \frac{1}{T} \int_0^T h(\xi_{-t})\, dt = \mathrm{E}[h(G_0(X))] \tag{36}$$

*If in addition the function $h$ is continuous, then (36) holds for each initial condition, and convergence is uniform in the initial phase angles.*

*Proof.* The result (36) for a.e. initial condition $\Phi_0$ is immediate from (35), the use of $G_0$ to define the probing signal in (9), and the definition $G(z) := G_0((z + z^{-1})/2)$.

To prove the stronger result for continuous $h$, we make explicit the dependency of the average on the initial condition:

$$h_t(\Phi_0) := h(G(e^{Wt}\Phi_0)) = h(\xi_t), \qquad \overline{h}_T(\Phi_0) := \frac{1}{T} \int_0^T h_t(\Phi_0)\, dt$$

We show that $\{h_t : t > 0\}$ and $\{\overline{h}_t : t > 0\}$ are each equicontinuous families of functions on $\Omega$: Since $e^{Wt}$ is an isometry on $\mathbb{C}^K$ for any $t$, there exists a $B_v < \infty$ such that:

$$\|G(e^{Wt}z) - G(e^{Wt}z')\| < B_v \|z - z'\|, \quad \text{for } z, z' \in \Omega \text{ and } t \geq 0 \tag{37}$$

Now, $h$ is uniformly continuous since this its domain is compact (we can take its domain to be the range of $G$). Consequently, for each $\varepsilon > 0$, there exists $\delta > 0$ such that for $x, x'$ in the domain of $h$,

$$\|h(x) - h(x')\| < \delta, \quad \text{if } \|x - x'\| < \varepsilon$$

Thus, by (37),

$$\|h_t(z) - h_t(z')\| < \delta, \quad \text{if } \|z - z'\| < \varepsilon/B_v$$

Equicontinuity of $\{h_t : t > 0\}$ and $\{\overline{h}_t : t > 0\}$ on $\Omega$ follows from these bounds. Pointwise convergence of $\overline{h}_T$ to $\langle h \rangle$ for a.e. $\Phi_0 \in \Omega$, as $T \to \infty$, then implies convergence from each initial condition, and also uniform convergence:

$$\lim_{T \to \infty} \max_{\Phi_0 \in \Omega} \|\overline{h}_T(\Phi_0) - \langle h \rangle\| = 0$$

$\square$

This proposition will be refined in the following.

If $g \colon \mathbb{R}^K \to \mathbb{R}$ is analytic in a neighborhood of $z \in \mathbb{R}^K$, we denote the mixed partials by

$$g^{(\alpha)}(z) = \frac{\partial^{\alpha_1}}{\partial z_1^{\alpha_1}} \cdots \frac{\partial^{\alpha_K}}{\partial z_K^{\alpha_K}} g(z), \quad \alpha \in \mathbb{Z}_+^K.$$

Denote $z^\alpha = \prod_i z_i^{\alpha_i}$, and $\alpha! = \prod_i \alpha_i!$ for $\alpha \in \mathbb{Z}_+^K$ with $0! := 1$. This notation is used to express the multivariate Taylor series formula in the following:

**Lemma B.3.** *Suppose that $g \colon \mathbb{R}^K \to \mathbb{R}$ is analytic in a neighborhood $\mathcal{N}_g$ of the hypercube $[-1, 1]^K$. Then there is $r_g > 1$ such that whenever $z \in [-1, 1]^K$ and $0 \leq r \leq r_g$ we have*

$$g(rz) = g(0) + \sum_{n=1}^\infty r^n \sum_{|\alpha|=n} \frac{1}{\alpha!} z^\alpha g^{(\alpha)}(0) \tag{38a}$$

*where the sum converges absolutely:* $\sum_{n=1}^\infty r_g^n \sum_{|\alpha|=n} \frac{1}{\alpha!} |g^{(\alpha)}(0)| < \infty$ $\tag{38b}$

A useful representation requires more notation:

$$g_0(\xi_t^0) := h(\xi_t) = h(G_0(\xi_t^0)), \quad t \geq 0, \tag{39}$$

with $\xi_{t,i}^0 = \cos(2\pi[\omega_i t + \phi_i])$ as defined by (9).

It follows that $g_0$ is analytic if both $h$ and $G_0$ also are. Along with this new notation, Lemma B.3 provides a representation of both $h(\xi_t)$ and the integral $\tilde{h}(\xi_t) = h(\xi_t) - \overline{h}$.

For $\alpha \in \mathbb{Z}_+^K$ we denote

$$\xi_t^\alpha = \prod_{i=1}^K \cos(2\pi[\omega_i t + \phi_i])^{\alpha_i} = 2^{-|\alpha|} \prod_{i=1}^K (\Phi_t^i + \Phi_t^{i*})^{\alpha_i}$$

$$\overline{\xi}^\alpha = \langle \xi^\alpha \rangle, \qquad \widetilde{\xi}_t^\alpha = \xi_t^\alpha - \overline{\xi}^\alpha \tag{40}$$

where $|\alpha| = \sum \alpha_i$ (the $\ell_1$-norm), and $\Phi_t^{i^*}$ denotes the complex conjugate. Let $B$ denote the set of all $K$-dimensional row vectors with entries in $\{-1, 1\}$. For fixed $\alpha \in \mathbb{Z}_+^K$, $|\alpha| \neq 0$, we decompose $b \in B$ as follows: $b = (b^1, b^2, \ldots, b^K)$, where $b^i$ has length $\alpha_i$ for each $i$, and necessarily has entries in $\{-1, 1\}$. These are used to define the frequency and phase variables

$$\omega_{\alpha,b} = \sum_{i=1}^{K} \alpha_i \omega_i \sum_k b_k^i \qquad \phi_{\alpha,b} = \sum_{i=1}^{K} \alpha_i \phi_i \sum_k b_k^i \tag{41}$$

Let $B_0^\alpha \subset B$ denote the set of vectors $b \in B$ for which $\omega_{\alpha,b} = 0$. Under the assumption that the frequencies are linearly independent over the rationals, this is equivalent to the following requirement:

$$\alpha_i \sum_k b_k^i = 0 \quad \text{for each } 1 \leq i \leq K$$

**Lemma B.4.** *The signal $\{\xi_t^\alpha\}$ in (40), its mean, and its centered integral admit the representations,*

$$\xi_t^\alpha = 2^{-|\alpha|} \sum_{b \in B^\alpha} \xi_t^{\alpha,b}, \qquad\qquad \overline{\xi^\alpha} = 2^{-|\alpha|} \sum_{b \in B_0^\alpha} \exp(j\phi_{\alpha,b})$$

$$\xi_t^{\alpha \mathrm{I}} = 2^{-|\alpha|} \sum_{b \notin B_0^\alpha} \frac{1}{2\pi j \omega_{\alpha,b}} \xi_t^{\alpha,b}, \qquad \xi_t^{\alpha,b} = \exp(2\pi j[\omega_{\alpha,b} t + \phi_{\alpha,b}]) \tag{42}$$

*Proof.* The representation for $\xi_t^\alpha$ is purely a change of notation. We have $\xi_t^{\alpha,b} = \exp(j\phi_{\alpha,b})$ (independent of $t$) when $b \in B_0$, and $\langle \xi^{\alpha,b} \rangle = 0$ otherwise. Consequently,

$$\overline{\xi^\alpha} = 2^{-|\alpha|} \sum_{b \in B^\alpha} \langle \xi^{\alpha,b} \rangle = 2^{-|\alpha|} \sum_{b \in B_0^\alpha} \exp(j\phi_{\alpha,b})$$

$$\xi_t^{\alpha \mathrm{I}} = \int_0^t [\xi_r^\alpha - \overline{\xi^\alpha}]\, dr - \overline{\xi}_t^{\alpha \mathrm{I}} = 2^{-|\alpha|} \sum_{b \notin B_0^\alpha} \int_0^t \xi_r^{\alpha,b}\, dr - \overline{\xi}_t^{\alpha \mathrm{I}} = 2^{-|\alpha|} \sum_{b \notin B_0^\alpha} \frac{1}{2\pi j \omega_{\alpha,b}} \xi_t^{\alpha,b}$$

$\square$

Before we can state the main result of this subsection we require a few more definitions. Let $B_+^\alpha \subset B$ denote the set of vectors $b \in B$ for which $\omega_{\alpha,b} > 0$, and $B_-^\alpha = \{-b : b \in B_+^\alpha\}$. We also require the following extension of the notation in (42):

$$z^{\alpha,b} := z_1^{n_1^{\alpha,b}} \cdots z_K^{n_K^{\alpha,b}}, \quad \text{with } n_i^{\alpha,b} = \alpha_i \sum_{k=1}^{\alpha_i} b_k^i, \quad z \in \mathbb{C}^K \setminus \{0\}$$

where the origin is avoided because $n_i^{\alpha,b} < 0$ for some $(i, \alpha, b)$. The following properties will be useful:

$$z^{\alpha,-b} = 1/z^{\alpha,b} \quad \text{and hence} \quad z^{\alpha,-b} = z^{\alpha,b^*} \quad \text{whenever } z \in \Omega. \tag{43}$$

where the star denotes complex conjugate.

**Theorem B.5.** *Suppose that the function $h \colon \mathbb{R}^m \to \mathbb{R}$ is analytic in a neighborhood $\mathcal{N}_h$ of the hypercube $[-1, 1]^m$, and suppose that $\xi_t$ is the $m$-dimensional probing signal (9) satisfying (QSA0). The following conclusions then hold:*

$$\tilde{h}(\xi_t) := h(\xi_t) - \overline{h} = \sum_{n=1}^{\infty} \sum_{|\alpha|=n} \frac{1}{\alpha!} g_0^{(\alpha)}(0) 2^{-|\alpha|} \sum_{b \in B_+^\alpha} 2\cos(2\pi[\omega_{\alpha,b} t + \phi_{\alpha,b}]) \tag{44a}$$

$$\int_{t_0}^{t_1} \tilde{h}(\xi_t)\, dt = \hat{h}(\Phi_{t_1}) - \hat{h}(\Phi_{t_0}), \quad 0 \leq t_0 \leq t_1 < \infty, \tag{44b}$$

$$\text{where} \quad \hat{h}(z) = -\frac{1}{2\pi} \sum_{n=1}^{\infty} \sum_{|\alpha|=n} \frac{1}{\alpha!} g_0^{(\alpha)}(0) 2^{-|\alpha|} \sum_{b \in B_+^\alpha} \frac{1}{j\omega_{\alpha,b}} [z^{\alpha,b} - z^{\alpha,-b}] \tag{44c}$$

*where $g_0 \colon \mathbb{R}^K \to \mathbb{R}$ is given by (39).*

*Moreover, the function $\hat{h}$ is analytic in the domain $\{z \in \mathbb{C}^K : 0 < \|z\| < r_{g_0}\}$, and admits the following representation when restricted to $\Omega$:*

$$\hat{h}(\Phi_t) = -\frac{1}{\pi} \sum_{n=1}^{\infty} \sum_{|\alpha|=n} \frac{1}{\alpha!} g_0^{(\alpha)}(0) 2^{-|\alpha|} \sum_{b \in B_+^\alpha} \frac{1}{\omega_{\alpha,b}} \sin(2\pi[\omega_{\alpha,b} t + \phi_{\alpha,b}]) \qquad (44d)$$

As mentioned after (12), the function $\hat{h}$ solving (44b) is known as the solution to Poisson's equation with forcing function $h$. This terminology is standard in ergodic theory for Markov processes. Since $\hat{h}$ is also analytic we are assured of multiple integrals that are also bounded in time:

$$\hat{h}(\Phi_T) = -\int_0^T \tilde{h}(\xi_t)\, dt + \hat{h}(\Phi_0)\,, \qquad \hat{\hat{h}}(\Phi_T) = -\int_0^T \hat{h}(\Phi_t) dt + \hat{\hat{h}}(\Phi_0) \qquad (44e)$$

Both are required in the analysis supporting our main results: recall the functions (17) used in Thm. 2.1 to define the terms in the p-mean flow representation.

The following corollary will prove useful. Note that we relax the assumption of analyticity.

**Corollary B.6.** *Suppose that (QSA0) holds, and that $g, h\colon \mathbb{R}^m \to \mathbb{R}$ are continuous functions. Suppose moreover that there is a zero-mean solution to Poisson's equation $\hat{h}$, solving (44b) for any $\Phi_0 \in \Omega$, and any $0 \le t_0 \le t_1 < \infty$. Then, $\langle g(\xi), \hat{h}(\Phi)\rangle = 0$.*

The proof of the theorem and its corollary are postponed to the end of this subsection.

It is clear from (44c) that we require a lower bound on $|\omega_{\alpha,b}|$ for $b \in B_+^\alpha$ in order to justify that $\hat{h}$ is analytic in a neighborhood of $\Omega$. Useful bounds are possible through application of extensions of Baker's Theorem concerning linear independence of algebraic numbers [28, 8].

The assumption that the $\{\omega_i\}$ defined in (32a) are linearly independent over the field of rational numbers is equivalent to the requirement that the rational numbers $\{r_i = a_i/b_i\}$ are *multiplicatively independent*. That is, for any integers $\{n_i : 1 \le i \le K\} \subset \mathbb{Z}$, the equation

$$r_1^{n_1} r_2^{n_2} \cdots r_K^{n_K} = 1$$

implies that $n_i = 0$ for each $i$. This is the language used in much of the literature surrounding Baker's Theorem. The following follows from [8, Thm. 1.8]:

**Theorem B.7.** *Under the assumptions of Thm. B.1 there is a constant $C > 0$ depending only on $\{a_i, b_i : 1 \le i \le K\}$ such that whenever $\omega_{\alpha,b} \neq 0$,*

$$|\omega_{\alpha,b}| \ge \beta_\alpha^{-C}\,, \qquad \beta_\alpha = \max\{3, \alpha_1, \ldots, \alpha_d\}$$

$\square$

Thm. B.1 follows immediately from Thm. B.5, for which the proof is given next:

*Proof of Thm. B.5.* The function $G_0$ is assumed analytic on $\mathbb{R}^K$ under (QSA0). Since $h$ is also analytic, this implies $g_0$ is analytic in a neighborhood $\mathcal{N}_{g_0}$ of the hypercube $[-1, 1]^K$.

The Taylor series expansion in Lemma B.3 combined with Lemma B.4 gives

$$h(\xi_t) = g_0(0) + \sum_{n=1}^{\infty} \sum_{|\alpha|=n} \frac{1}{\alpha!} g_0^{(\alpha)}(0) \xi_t^\alpha$$

$$= g_0(0) + \sum_{n=1}^{\infty} \sum_{|\alpha|=n} \frac{1}{\alpha!} g_0^{(\alpha)}(0) 2^{-|\alpha|} \sum_{b \in B^\alpha} \xi_t^{\alpha,b} \qquad (45)$$

Obtaining the mean of each side gives

$$\bar{h} = g_0(0) + \sum_{n=1}^{\infty} \sum_{|\alpha|=n} \frac{1}{\alpha!} g_0^{(\alpha)}(0) 2^{-|\alpha|} \sum_{b \in B_0^\alpha} \xi_t^{\alpha,b}$$

where $\xi_t^{\alpha,b} = \exp(2\pi j\phi_{\alpha,b})$ for $b \in B_0^\alpha$; similar arguments were used in the derivation of (42). Subtracting $\overline{h}$ from each side of (45) gives

$$\tilde{h}(\xi_t) = \sum_{n=1}^\infty \sum_{|\alpha|=n} \frac{1}{\alpha!} g_0^{(\alpha)}(0) 2^{-|\alpha|} \sum_{b \notin B_0^\alpha} \xi_t^{\alpha,b} \tag{46}$$

The proof of (44a) is completed on observing that

$$\sum_{b \notin B_0^\alpha} \xi_t^{\alpha,b} = \sum_{b \in B_+^\alpha} \xi_t^{\alpha,b} + \sum_{b \in B_-^\alpha} \xi_t^{\alpha,b} = \sum_{b \in B_+^\alpha} [\xi_t^{\alpha,b} + \xi_t^{\alpha,-b}] = 2\cos(2\pi[\omega_{\alpha,b} t + \phi_{\alpha,b}])$$

The representation (46) for $\tilde{h}$ motivates the following definition:

$$\hat{h}(\Phi_t) := -\sum_{n=1}^\infty \sum_{|\alpha|=n} \frac{1}{\alpha!} g_0^{(\alpha)}(0) \xi_t^{\alpha I} \tag{47}$$

whose extension to $\mathbb{C}^K$ given in (44c) is duplicated here for convenience:

$$\hat{h}(z) = -\frac{1}{2\pi} \sum_{n=1}^\infty \sum_{|\alpha|=n} \frac{1}{\alpha!} g_0^{(\alpha)}(0) 2^{-|\alpha|} \sum_{b \in B_+^\alpha} \frac{1}{j\omega_{\alpha,b}} [z^{\alpha,b} - z^{\alpha,-b}]$$

The extension follows from the preceding arguments, using $\omega_{\alpha,-b} = -\omega_{\alpha,b}$.

The remainder of the proof consists of two parts: show that $\hat{h}$ is analytic in the region $\{z \in \mathbb{C}^K : 0 < \|z\| < r_{g_0}\}$, and then establish the desired properties when $z \in \Omega$. Those desired properties are firstly $\frac{d}{dt}\hat{h}(\Phi_t) = -\tilde{h}(\xi_t)$, which follows from (44a) provided the sum in (47) converges absolutely. The final property is the representation (44d) in terms of sums of $\sin(2\pi[\omega_{\alpha,b} t + \phi_{\alpha,b}])$. This is also immediate since

$$\tfrac{1}{j}[z^{\alpha,b} - z^{\alpha,-b}] = 2\sin(2\pi[\omega_{\alpha,b} t + \phi_{\alpha,b}]) \quad \text{when } z = \Phi_t$$

To complete the proof we establish that $\hat{h}$ is analytic in the given domain.

From (44c) it follows that $\hat{h}$ is a function of the $2K$-dimensional vector valued function $v(z) = (z_1, \dots, z_K, z_1^{-1}, \dots, z_K^{-1})$. The mapping $z \mapsto v(z)$ is analytic in $\mathbb{C}^K \setminus \{0\}$, so it suffices to obtain the bound

$$B_h(r) := \sum_{n=1}^\infty \sum_{|\alpha|=n} \frac{1}{\alpha!} |g_0^{(\alpha)}(0)| 2^{-|\alpha|} \sum_{b \in B_+^\alpha} \omega_{\alpha,b}^{-1} < \infty, \qquad \text{for all } r < r_{g_0}.$$

It will follow that $\hat{h}$ is analytic in the set $\{z \in \mathbb{C}^K : 0 < \|z\| < r_{g_0}\}$.

Thm. B.7 gives the bound $\omega_{\alpha,b}^{-1} \le 3^C + n^C$ with $n = |\alpha|$. Consequently, for any $r > 0$,

$$B_h(r) \le \sum_{n=1}^\infty r^n [3^C + n^C] \sum_{|\alpha|=n} \frac{1}{\alpha!} |g_0^{(\alpha)}(0)| \tag{48}$$

The right hand side is finite for any $r < r_{g_0}$ due to the bound (38b). $\qquad \square$

We turn next to the proof of Corollary B.6, which will follow from a sequence of lemmas.

**Lemma B.8.** *Suppose that the functions $g, h : \mathbb{R}^m \to \mathbb{R}$ are analytic in a neighborhood $\mathcal{N}_h$ of the hypercube $[-1,1]^m$, and suppose that $\xi_t$ is the $m$-dimensional probing signal (9) satisfying (QSA0). Then, $\langle g(\xi), \hat{h}(\Phi) \rangle = 0$, where $\hat{h}$ is given in (44c).*

*Proof.* Given the representation (44d) it suffices to show that $\langle \gamma, \zeta \rangle = 0$ for the functions defined by $\gamma_t = \cos(2\pi[\omega_{\alpha,b} t + \phi_{\alpha,b}])$ and $\zeta_t = \sin(2\pi[\omega_{\alpha',b'} t + \phi_{\alpha',b'}])$, with $(\alpha, b)$, $(\alpha', b')$ arbitrary pairs appearing in the sum that represents $\hat{h}$.

If $\omega_{\alpha,b} \neq \omega_{\alpha',b'}$ then the conclusion $\langle \gamma, \zeta \rangle = 0$ is immediate (including the case $\omega_{\alpha,b} = 0$).

If $\omega_{\alpha,b} = \omega_{\alpha',b'}$ it follows from the definition (41) that $\phi_{\alpha,b} = \phi_{\alpha',b'}$, and the conclusion $\langle \gamma, \zeta \rangle = 0$ follows from the double angle identity:

$$\gamma_t \zeta_t = \sin(4\pi[\omega_{\alpha,b}t + \phi_{\alpha,b}])/2$$

□

The next result is required for approximating $h$ and $\hat{h}$ simultaneously by analytic functions. Denote for $\varepsilon > 0$,

$$\hat{h}^\varepsilon(z) = \int_0^\infty e^{-\varepsilon t} \tilde{h}(e^{Wt} z)\, dt, \quad z \in \Omega \tag{49}$$

**Lemma B.9.** *Under the assumptions of Corollary B.6 we have*

$$\hat{h}^\varepsilon(z) = \hat{h}(z) - \varepsilon \int_0^\infty e^{-\varepsilon t} \hat{h}(e^{Wt} z)\, dt$$

$$\lim_{\varepsilon \downarrow 0} \max_{z \in \Omega} |\hat{h}^\varepsilon(z) - \hat{h}(z)| = 0$$

*Proof.* The first limit follows from the differential representation of the solution to Poisson's equation: $\hat{h}(e^{Wt} z)$ is absolutely continuous, with $d\hat{h}(e^{Wt} z) = -\tilde{h}(e^{Wt} z)dt$.

The second limit then follows from the assumption that the mean of $\hat{h}$ is zero. □

*Proof of Corollary B.6.* Lemma B.8 covers the special case in which $g, h$ are analytic.

Consider next the case in which $g$ is analytic, but $h$ and $\hat{h}$ are only assumed continuous. Let $\varepsilon > 0$, $n \geq 1$ be arbitrary, and apply the Stone-Weierstrass Theorem to obtain a polynomial function $h_n$ satisfying $|h(x) - h_n(x)| \leq \varepsilon/n$ for all $x \in [-1,1]^m$. It follows from the definition (49) that $|\hat{h}^\varepsilon(z) - \hat{h}_n^\varepsilon(z)| \leq 1/n$ for all $z \in \Omega$.

We then have

$$\langle g(\xi), \hat{h}(\Phi) \rangle = \langle g(\xi), \hat{h}^\varepsilon(\Phi) \rangle + o(1)$$
$$= \langle g(\xi), \hat{h}_n^\varepsilon(\Phi) \rangle + o(1) + O(\|g\|_\infty/n)$$
$$= \langle g(\xi), \hat{h}_n(\Phi) \rangle + o(1) + O(\|g\|_\infty/n) + o_n(1)$$

with $\|g\|_\infty$ the maximum of $|g(x)|$ over $x \in [-1,1]^m$ and $\hat{h}_n$ is a polynomial function defined by Thm. B.5 using $h_n$. In the final approximation, $o_n(1) \to 0$ as $\varepsilon \downarrow 0$ for each fixed $n$, but the convergence is not necessarily uniform in $n$. Letting $\varepsilon \downarrow 0$ gives,

$$\langle g(\xi), \hat{h}(\Phi) \rangle = \langle g(\xi), \hat{h}_n(\Phi) \rangle + O(\|g\|_\infty/n) = O(\|g\|_\infty/n)$$

where the second bound holds because $g$ and $h_n$ satisfy the assumptions of Lemma B.8. It follows that $\langle g, \hat{h} \rangle = 0$ since $n \geq 1$ is arbitrary.

The general case is similar but simpler: apply the Stone-Weierstrass Theorem to obtain a polynomial function $g_n$ satisfying $|g(x) - g_n(x)| \leq 1/n$ for all $x \in [-1,1]^m$. From the previous bound we have

$$\langle g(\xi), \hat{h}(\Phi) \rangle = \langle g_n(\xi), \hat{h}(\Phi) \rangle + O(1/n) = O(1/n)$$

This completes the proof, since $n \geq 1$ is arbitrary. □

## B.2 Extensions to Discrete Time

The extension of Thm. B.1 to the discrete time setting is essentially unchanged, though obtaining bounds on the constants is more challenging [50, 8].

**Theorem B.10.** *Suppose that the assumptions of Thm. B.1 hold. Then there is a finite constant $B_f$ independent of the phase values $\{\phi_i\}$ such that*

$$\left| \frac{1}{N} \sum_{k=1}^N \tilde{h}(\xi_k) \right| \leq B_f \frac{1}{N}, \quad N \geq 1 \tag{50}$$

□

We will see in the proof that the discrete time case is more complex because we require a stronger condition on $\omega_{\alpha,b}$. A useful bound is obtained from [8, Thm. 2.1]:

**Theorem B.11.** *Under the assumptions of Thm. B.1 there are constants $\delta, C > 0$ depending only on $\{a_i, b_i : 1 \le i \le K\}$ such that whenever $\omega_{\alpha,b} \neq 0$ and $n_0 \in \mathbb{Z}$,*

$$|n_0 + \omega_{\alpha,b}| \ge \delta\beta_\alpha^{-C}, \quad \beta_\alpha = \max\{3, \alpha_1, \dots, \alpha_d\}$$

$\square$

*Proof of Thm. B.10.* Denote the partial sums,

$$S_N = \sum_{k=0}^{N} \tilde{h}(\xi_k)$$

Motivated by the foregoing, to bound the sum we consider sums of the primitives obtained from the Taylor series expansion (38a):

$$S_N^{\alpha,b} := \sum_{k=0}^{N} \exp(2\pi j[\omega_{\alpha,b}k + \phi_{\alpha,b}])$$

$$S_N^{\alpha} := \sum_{b \notin B_0^\alpha} S_N^{\alpha,b}$$

The expansion (38a) tells us that $S_N$ admits the representation

$$S_N = \sum_{n=1}^{\infty} \sum_{|\alpha|=n} \frac{1}{\alpha!} g_0^{(\alpha)}(0) S_N^\alpha$$

To prove the theorem, it suffices to establish a uniform bound over $N$ and $\{\phi_i\}$.

Consider any $\alpha$ and $b \notin B_0^\alpha$. On denoting $w = \exp(2\pi j\omega_{\alpha,b})$ we obtain

$$S_N^{\alpha,b} = \exp(2\pi j\phi_{\alpha,b}) \sum_{k=0}^{N} w^k = \exp(2\pi j\phi_{\alpha,b}) \frac{w^{N+1} - 1}{w - 1}, \qquad |S_N^{\alpha,b}| \le 2\frac{1}{|w - 1|}$$

Given $|w - 1|^2 = |1 - \cos(2\pi\omega_{\alpha,b})|^2 + \sin(2\pi\omega_{\alpha,b})^2$ it is not enough to bound $|\omega_{\alpha,b}|$ from zero as in the continuous time case. Rather, we require a bound on $\inf_{n_0 \in \mathbb{Z}} |n_0 + \omega_{\alpha,b}|$. Thm. B.11 gives us the desired bound: for some $\delta_0 > 0$ and any $n_0 \in \mathbb{Z}$,

$$\inf_{n_0 \in \mathbb{Z}} |n_0 + \omega_{\alpha,b}| \ge \delta_0\beta_\alpha^{-C}$$

The remainder of the proof that $\{S_N : N \ge 1\}$ is bounded is identical to the continuous time case (see arguments surrounding (48)). $\square$

## B.3 The Perturbative Mean Flow Representation

The derivation of the perturbative mean flow representation (5) is based upon the solutions to Poisson's equation in (17). We start by re-writing the QSA ODE (3) in terms of the *apparent noise* $\{\widetilde{\Xi}_t\}$,

$$\tfrac{d}{dt}\Theta_t = a_t[\bar{f}(\Theta_t) + \widetilde{\Xi}_t], \qquad \widetilde{\Xi}_t := f(\Theta_t, \xi_t) - \bar{f}(\Theta_t)$$

The proof of the first part of Thm. 2.1 follows directly from the next three lemmas.

**Lemma B.12.** *Under (QSA5), the apparent noise $\widetilde{\Xi}_t$ can be expressed as*

$$\widetilde{\Xi}_t = -\tfrac{d}{dt}\hat{f}(\Theta_t, \Phi_t) - a_t\Upsilon(\Theta_t, \Phi_t) \tag{51}$$

*Proof.* Applying (31) with $h = \hat{f}$,

$$\tfrac{d}{dt}\hat{f}(\Theta_t, \Phi_t) = \partial_\theta\hat{f}(\Theta_t, \Phi_t)\tfrac{d}{dt}\Theta_t - [f(\Theta_t, \xi_t) - \bar{f}(\Theta_t)]$$

This gives

$$\widetilde{\Xi}_t = -\tfrac{d}{dt}\hat{f}(\Theta_t, \Phi_t) + a_t\partial_\theta\hat{f}(\Theta_t, \Phi_t)f(\Theta_t, \xi_t)$$

Finally, (51) follows from (18). $\square$

**Lemma B.13.** *Suppose that (QSA1) and (QSA5) hold. If $a_t = (1+t)^{-\rho}$, with $\rho \in (0,1)$,*

$$\tfrac{d}{dt}\hat{f}(\Theta_t, \Phi_t) = -r_t a_t [\mathcal{D}^f \hat{\hat{f}}](\Theta_t, \Phi_t) + a_t \tfrac{d}{dt}[\mathcal{D}^f \hat{\hat{f}}](\Theta_t, \Phi_t) - \tfrac{d^2}{dt^2}\hat{\hat{f}}(\Theta_t, \Phi_t) \tag{52}$$

*where $r_t = \rho/(t+1)$.*

*Proof.* Similarly to Lemma B.12, applying (31) with $h = \hat{\hat{f}}$ gives

$$\tfrac{d}{dt}\hat{f}(\Theta_t, \Phi_t) = a_t [\mathcal{D}^f \hat{\hat{f}}](\Theta_t, \Phi_t) - \hat{f}(\Theta_t, \Phi_t)$$

Differentiating both sides with respect to $t$ yields

$$\begin{aligned}
\tfrac{d^2}{dt^2}\hat{f}(\Theta_t, \Phi_t) &= \tfrac{d}{dt}\{a_t [\mathcal{D}^f \hat{\hat{f}}](\Theta_t, \Phi_t)\} - \tfrac{d}{dt}\hat{f}(\Theta_t, \Phi_t) \\
&= -r_t a_t [\mathcal{D}^f \hat{\hat{f}}](\Theta_t, \Phi_t) + a_t \tfrac{d}{dt}[\mathcal{D}^f \hat{\hat{f}}](\Theta_t, \Phi_t) - \tfrac{d}{dt}\hat{f}(\Theta_t, \Phi_t)
\end{aligned}$$

where the last equality follows from the product rule. The result in (52) is then achieved upon rearranging terms. $\square$

**Lemma B.14.** *Under (QSA5),*

$$\Upsilon(\Theta_t, \Phi_t) = \bar{\Upsilon}(\Theta_t) + a_t [\mathcal{D}^f \widehat{\Upsilon}](\Theta_t, \Phi_t) - \tfrac{d}{dt}\widehat{\Upsilon}(\Theta_t, \Phi_t) \tag{53}$$

*Proof.* Again, applying (31) with $h = \widehat{\Upsilon}$,

$$\tfrac{d}{dt}\widehat{\Upsilon}(\Theta_t, \Phi_t) = a_t [\mathcal{D}^f \widehat{\Upsilon}](\Theta_t, \Phi_t) - [\Upsilon(\Theta_t, \Phi_t) - \bar{\Upsilon}(\Theta_t)]$$

which gives (53) after rearranging terms. $\square$

We conclude this subsection with the remainder of the proof of Thm. 2.1 (i). Proofs of the remaining parts are given after the theorem statement: Thm. 2.1 (ii) follows from Corollary B.6 and the representation $\Upsilon_i(\theta, \Phi) = -\sum_{j=1}^{d} \widehat{A}_{i,j}(\theta, \Phi) f_j(\theta, \xi)$, and (iii) follows from [30, Prop. 4.33 and 4.34].

*Proof of part (i) in Thm. 2.1.* This part contains the details of the p-mean flow representation (19). It is obtained upon substitution of (52) and (53) into (51). $\square$

## B.4  Acceleration

We turn next to analysis of PR and FB techniques.

**Tighter bounds for PR averaging**  As explained in Section 2.1, convergence of the QSA ODE is established by the coupling of $\{\Theta_t\}$ and $\{\bar{\Theta}_t\}$ for $t \geq t_0$, for some $t_0$ that depends on the stability properties of $\dot{x} = \bar{f}(x)$. This coupling is used to establish boundedness of the scaled error,

$$Z_t := \frac{\Theta_t - \bar{\Theta}_t}{a_t}, \quad t \geq t_0 \tag{54}$$

Convergence of $\{\bar{\Theta}_t\}$ to $\theta^*$ is typically very fast when $a_t = (1+t)^{-\rho}$ and $\rho < 1$:

**Lemma B.15.** *Suppose (QSA1) – (QSA4) hold. If $a_t = (1+t)^{-\rho}$, with $0 < \rho < 1$, then*

$$Z_t = \frac{\Theta_t - \theta^*}{a_t} + \varepsilon_t^{\bar{\Theta}}$$

*with $\varepsilon_t^{\bar{\Theta}} = [\theta^* - \bar{\Theta}_t]/a_t$ vanishing faster than $O(T^{-N})$ for any $N \geq 1$.* $\square$

This lemma is part of [30, Prop. 4.26]. Combined with (21a) it gives

$$Z_t = \bar{Y}^* - \hat{f}_t^* + O(a_t), \quad t \geq t_0 \tag{55}$$

The next result is of [30, Thm. 4.25]. It is a primitive version of the p-mean flow representation that will serve as a foundation for the proof of Thm. 2.2:

**Theorem B.16.** *Suppose the assumptions of Thm. 2.2 hold, then*

$$\Theta_T^{\mathsf{PR}} = \theta^* + a_T c(\rho, \kappa)\bar{Y}^* + \mathcal{B}_T/T \tag{56}$$

*where $\{\mathcal{B}_T\}$ is bounded in $T$ and $c(\rho, \kappa)$ is defined in Thm. 2.2 .* □

The key step in the proof of Thm. 2.2 is to bound the process $\{\mathcal{B}_T\}$ in (56). It is expressed in [30, Thm. 4.25] as

$$\mathcal{B}_T = [A^*]^{-1}\{\varepsilon_T^Y - \varepsilon_T^Z + \varepsilon_T^\Upsilon + \varepsilon_T^a\} \tag{57}$$

where for $Y_t := Z_t + \hat{f}(\Theta_t, \Phi_t)$ (recall (32e)), $r_t = \rho/(1+t)$ and $\widetilde{\Upsilon}_t^* = \Upsilon_t^* - \bar{\Upsilon}^*$ (recall (17) and (18)),

$$\varepsilon_T^Y = Y_T - Y_{T0}, \qquad \varepsilon_T^Z = \int_{T_0}^T r_t Z_t \, dt, \qquad \varepsilon_T^\Upsilon = \int_{T_0}^T a_t \widetilde{\Upsilon}_t^* \, dt, \qquad \varepsilon_T^a = \int_{T_0}^T a_t O(\|\Theta_t - \theta^*\|) \, dt$$

We proceed to bound each term.

**Lemma B.17.** *Under the assumptions of Thm. 2.2, $\varepsilon_T^Y = O(a_T)$.*

*Proof.* By substitution of (55) into the definition of $Y_t$,

$$Y_t = \bar{Y}^* - \hat{f}_t^* + \hat{f}(\Theta_t, \Phi_t) + O(a_t)$$
$$= \bar{Y}^* + O(\|\Theta_t - \theta^*\|) + O(a_t) = \bar{Y}^* + O(a_t)$$

where the last equalities follow from Lipschitz continuity of $\hat{f}_t(\Theta_t, \Phi_t)$ together with the fact that $\|\Theta_t - \theta^*\| = O(a_t)$ from Thm. 2.2. Now, $a_{T_0} = O(a_T)$ since $T_0 = (1 - 1/\kappa)T$ and hence

$$\varepsilon_T^Y = Y_T - Y_{T0} = O(a_T)$$

□

**Lemma B.18.** *Under the assumptions of Thm. 2.2,*

$$\varepsilon_T^Z = \varepsilon_\infty^Z + O(a_T), \qquad \varepsilon_\infty^Z = \bar{Y}^* \log(\kappa/(\kappa - 1))\rho$$

*Proof.* For $r_t$ as defined below (57) and $a_t, T_0$ as defined by Thm. 2.2, we have the following:

$$\int_{T_0}^T r_t \, dt = \rho \log(\kappa/(\kappa - 1)), \qquad \int_{T_0}^T r_t O(a_t) \, dt = O(a_T)$$

The above identities along with the representation of $Z_t$ in (55) imply

$$\varepsilon_T^Z = \int_{T_0}^T r_t[\bar{Y}^* - \hat{f}_t^* + O(a_t)] \, dt = \bar{Y}^* \log(\kappa/(\kappa - 1))\rho + \int_{T_0}^T r_t \hat{f}_t^* \, dt + O(a_T) \tag{58}$$

It remains to bound the last integral in the right side of (58). By integration by parts,

$$\int_{T_0}^T r_t \hat{f}_t^* \, dt = \int_{T_0}^T r_t \, d\hat{\hat{f}}_t^* = r_T \hat{\hat{f}}_T^* - r_{T_0} \hat{\hat{f}}_{T_0}^* + \frac{1}{\rho} \int_{T_0}^T r_t^2 \hat{\hat{f}}_t^* \, dt$$

Here, $\hat{\hat{f}}_t^*$ is bounded by assumption in (QSA5) so we have the bound

$$\int_{T_0}^T r_t^2 \hat{\hat{f}}_t^* \, dt \leq \sup_t \|\hat{\hat{f}}_t^*\| \int_{T_0}^T r_t^2 \, dt = O(r_T)$$

Thus, $\int_{T_0}^T r_t \hat{f}_t^* \, dt = O(r_T)$. The result then follows from substitution of this bound into (58). □

**Lemma B.19.** *Under the assumptions of Thm. 2.2, $\|\varepsilon_T^\Upsilon\| = O(a_T)$.*

*Proof.* Again applying integration by parts,

$$\varepsilon_T^\Upsilon = -\int_{T_0}^T a_t \, d\widehat{\Upsilon}_t^* = a_T \widehat{\Upsilon}_T^* - a_{T_0} \widehat{\Upsilon}_{T_0}^* + \int_{T_0}^T r_t a_t \widehat{\Upsilon}_t^* \, dt \tag{59}$$

where $\widehat{\Upsilon}_t^*$ is as defined by (32i). We have that $\widehat{\Upsilon}_t^*$ is bounded by assumption in (QSA5) and we have the bound

$$\int_{T_0}^T r_t a_t \widehat{\Upsilon}_t^* \, dt \le \sup_t \|\widehat{\Upsilon}_t^*\| \int_{T_0}^T r_t a_t \, dt = O(a_T)$$

Then, we obtain the desired conclusion by substitution of the above bound into (59). □

**Lemma B.20.** *Under the assumptions of Thm. 2.2, $\varepsilon_T^a = O(T^{1-2\rho})$.*

*Proof.* From Thm. 2.2, $\|\Theta_t - \theta^*\| = O(a_t)$. Thus,

$$\varepsilon_T^a = \int_{T_0}^T a_t O(\|\Theta_t - \theta^*\|) \, dt = \int_{T_0}^T O(a_t^2) \, dt = O(T^{1-2\rho})$$

□

*Proof of Thm. 2.2.* The proof of (21a) can be found in [30, Thm. 4.24].

Combining Lemmas B.17 to B.20:

$$\mathcal{B}_T = -[A^*]^{-1} \bar{Y}^* \log(\kappa/(\kappa-1))\rho + O(T^{1-2\rho})$$

Then, (21b) follows from substitution of the above representation into (56). □

**Forward-backward filtering** We refer to $\xi_t$ as the forward probing signal, and $\xi_t^-$ as backward probing. Similarly to $\xi_t^-$, we introduce the signal $\Phi_t^- := \Phi_{-t}$. The first step in establishing the identity $\bar{Y}^{*-} = -\bar{Y}^*$ is to show that the solutions to Poisson's equation (32f) for the two QSA ODEs differ by a negative sign.

**Lemma B.21.** *Suppose that (QSA5) holds, and let $\widehat{A}^*$ and $\widehat{A}^{*-}$ denote the solutions to Poisson's equation satisfying:*

$$\widehat{A}^*(\Phi_{t_0}) = \int_{t_0}^{t_1} \tilde{A}(\theta^*, \xi_t) \, dt + \widehat{A}^*(\Phi_{t_1})$$

$$\widehat{A}^{*-}(\Phi_{t_0}^-) = \int_{t_0}^{t_1} \tilde{A}(\theta^*, \xi_t^-) \, dt + \widehat{A}^{*-}(\Phi_{t_1}^-), \quad any \; 0 \le t_0 \le t_1,$$

*normalized so that $\langle \widehat{A}^* \rangle = \langle \widehat{A}^{*-} \rangle = 0$, as in (32g). We then have $\widehat{A}^*(z) = -\widehat{A}^{*-}(z)$ for $z \in \Omega$.*

*Proof.* Through a change of variables when $t_0 = 0$, $t_1 = t$,

$$\widehat{A}^-(\Phi_t^-) = \int_0^{-t} -\tilde{A}(\theta^*, \xi_\tau) \, d\tau + \widehat{A}^{*-}(\Phi_0^-)$$

Differentiating each side gives

$$\tfrac{d}{dt} \widehat{A}^{*-}(\Phi_t^-) = \tilde{A}(\theta^*, \xi_t^-) = -\tfrac{d}{dt} \widehat{A}^*(\Phi_t^-)$$

That is, $\widehat{A}^*(z) = -\widehat{A}^{*-}(z) + M^\circ$ for a constant matrix $M^\circ \in \mathbb{R}^{d \times d}$. The conclusion $M^\circ = 0$ follows from the assumed normalization on the means of $\widehat{A}^*$ and $\widehat{A}^{*-}$. □

Recall from (20) that $\bar{\Upsilon}^*$ and its analogous quantity $\bar{\Upsilon}^{*-}$ for the QSA ODE with backward probing (22) can be expressed

$$\bar{\Upsilon}^* = - \lim_{T \to \infty} \frac{1}{T} \int_0^T \widehat{A}^*(\Phi_t) f(\theta^*, \xi_t) \, dt \tag{60a}$$

$$\bar{\Upsilon}^{*-} = - \lim_{T \to \infty} \frac{1}{T} \int_0^T \widehat{A}^{*-}(\Phi_t^-) f(\theta^*, \xi_t^-) \, dt \tag{60b}$$

where $\widehat{A}^*$ and $\widehat{A}^{*-}$ are as defined by Lemma B.21. We now show that the application of Lemma B.21 and Prop. B.2 to (60a) and (60b) leads to the proof of Thm. 2.3.

*Proof of Thm. 2.3.* From Thm. 2.2, the following holds for $\rho \in (1/2, 1)$:

$$\Theta_T^{\mathsf{FB}} = \tfrac{1}{2}[\Theta_T^{\mathsf{PR}} + \Theta_T^{\mathsf{PR}^-}]$$
$$= \theta^* + \tfrac{1}{2} a_T c(\kappa, \rho)(\bar{Y}^* + \bar{Y}^{*-}) + \tfrac{1}{2}(\mathcal{B}_T + \mathcal{B}_T^-)/T \tag{61}$$

An application of the backwards in time LLN in Prop. B.2 with $h(\xi_t) = -\widehat{A}^*(\Phi_t) f(\theta^*, \xi_t)$ along with (60a) yields

$$\bar{\Upsilon}^* = - \lim_{T \to \infty} \frac{1}{T} \int_0^T \widehat{A}^*(\Phi_t^-) f(\theta^*, \xi_t^-) \, dt$$

Then, by Lemma B.21, we get that $\widehat{A}^*(\Phi_t^-) f(\theta^*, \xi_t^-) = -\widehat{A}^{*-}(\Phi_t^-) f(\theta^*, \xi_t^-)$. Together with (60b) we obtain

$$\bar{\Upsilon}^{*-} = - \lim_{T \to \infty} \frac{1}{T} \int_0^T \widehat{A}^{*-}(\Phi_t^-) f(\theta^*, \xi_t^-) \, dt = \lim_{T \to \infty} \frac{1}{T} \int_0^T \widehat{A}^*(\Phi_t^-) f(\theta^*, \xi_t^-) \, dt = -\bar{\Upsilon}^*$$

which implies $\bar{Y}^{*-} = -\bar{Y}^*$ via (20). Combining this result with (61) completes the proof. $\square$

## B.5 QSA applied to Gradient-Free Optimization

Recall that $\Gamma : \mathbb{R}^d \to \mathbb{R}$ is strongly convex if there exists a constant $\delta_0 > 0$ such that:

$$\delta_0 \|\theta - \theta^0\|^2 \leq (\nabla\Gamma(\theta) - \nabla\Gamma(\theta^0)) \cdot (\theta - \theta^0), \quad \forall \theta, \theta^0 \in \mathbb{R}^d \tag{62}$$

The bias bound in Section 3.1 is a corollary of the fact that 1qSGD and 2qSGD have identical average vector fields under mild assumptions on $\boldsymbol{\xi}$. This follows from Lemma B.22 and Prop. 3.1.

**Lemma B.22.** *Suppose $\boldsymbol{\xi}$ is defined using (10), with $\{\omega_i\}$ distinct. Then, for any constant matrix $M \in \mathbb{R}^{d \times d}$,*

$$\langle \boldsymbol{\xi} \boldsymbol{\xi}^{\mathsf{T}} M \boldsymbol{\xi} \rangle = 0$$

*Proof.* We apply Prop. B.2. The arcsine law used in the proposition is odd, meaning that $\boldsymbol{\xi}$ and $-\boldsymbol{\xi}$ have the same steady-state distribution, giving $\langle \boldsymbol{\xi} \boldsymbol{\xi}^{\mathsf{T}} M \boldsymbol{\xi} \rangle = -\langle \boldsymbol{\xi} \boldsymbol{\xi}^{\mathsf{T}} M \boldsymbol{\xi} \rangle = 0$. $\square$

*Proof of Prop. 3.1.* Following the proof of Lemma B.22, $\boldsymbol{\xi}$ and $-\boldsymbol{\xi}$ have the same steady-state distribution by Prop. B.2, which gives

$$\langle f^{1\mathsf{Q}}(\theta, \boldsymbol{\xi}) \rangle = -\frac{1}{\varepsilon} \langle \boldsymbol{\xi} \Gamma(\theta + \varepsilon \boldsymbol{\xi}) \rangle = \frac{1}{\varepsilon} \langle \boldsymbol{\xi} \Gamma(\theta - \varepsilon \boldsymbol{\xi}) \rangle = \langle f^{1\mathsf{Q}}(\theta, -\boldsymbol{\xi}) \rangle$$

This implies equality of the average vector fields for $f^{1\mathsf{Q}}$ and $f^{2\mathsf{Q}}$:

$$\bar{f}(\theta) = \langle f^{2\mathsf{Q}}(\theta, \boldsymbol{\xi}) \rangle = -\frac{1}{2\varepsilon} \langle \boldsymbol{\xi} \Gamma(\theta + \varepsilon \boldsymbol{\xi}) \rangle + \frac{1}{2\varepsilon} \langle \boldsymbol{\xi} \Gamma(\theta - \varepsilon \boldsymbol{\xi}) \rangle$$
$$= -\frac{2}{2\varepsilon} \langle \boldsymbol{\xi} \Gamma(\theta + \varepsilon \boldsymbol{\xi}) \rangle = \langle f^{1\mathsf{Q}}(\theta, \boldsymbol{\xi}) \rangle$$

Now, by a second order Taylor series expansion of $f^{1Q}(\theta, \xi_t)$ around $\theta$,

$$f^{1Q}(\theta, \xi_t) = -\frac{1}{\varepsilon}\xi_t\Gamma(\theta) - \xi_t\xi_t^\mathsf{T}\nabla\Gamma(\theta) - \frac{\varepsilon}{2}\xi_t\xi_t^\mathsf{T}\nabla^2\Gamma(\theta)\xi_t + O(\varepsilon^2)$$

Taking the mean of each side yields

$$\bar{f}(\theta) = -\Sigma_\xi\nabla\Gamma(\theta) - \frac{\varepsilon}{2}\langle\xi\xi^\mathsf{T}\nabla^2\Gamma(\theta)\xi\rangle + O(\varepsilon^2)$$

We have $\langle\xi\xi^\mathsf{T}\nabla^2\Gamma(\theta)\xi\rangle = 0$ by Lemma B.22, which gives (25). $\qquad\square$

The proof of Corollary 3.2 then follows from applying the results of Prop. 3.1 to (62).

*Proof of Corollary 3.2.* Since $\Gamma$ satisfies (62) and $\nabla\Gamma(\theta^{\mathsf{opt}}) = 0$, we achieve the following for $\delta > 0$:

$$\delta\|\theta^* - \theta^{\mathsf{opt}}\|^2 \leq (\nabla\Gamma(\theta^*) - \nabla\Gamma(\theta^{\mathsf{opt}})) \cdot (\theta^* - \theta^{\mathsf{opt}})$$
$$\delta\|\theta^* - \theta^{\mathsf{opt}}\|^2 \leq \|\nabla\Gamma(\theta^*)\| \, \|\theta^* - \theta^{\mathsf{opt}}\|$$

We have that $\bar{f}(\theta^*) = 0$ by (QSA3), which implies $\nabla\Gamma(\theta^*) = O(\varepsilon^2)$ from (25) under the assumption that $\Sigma_\xi > 0$. Thus,

$$\|\theta^* - \theta^{\mathsf{opt}}\| \leq O(\varepsilon^2)$$

$\qquad\square$

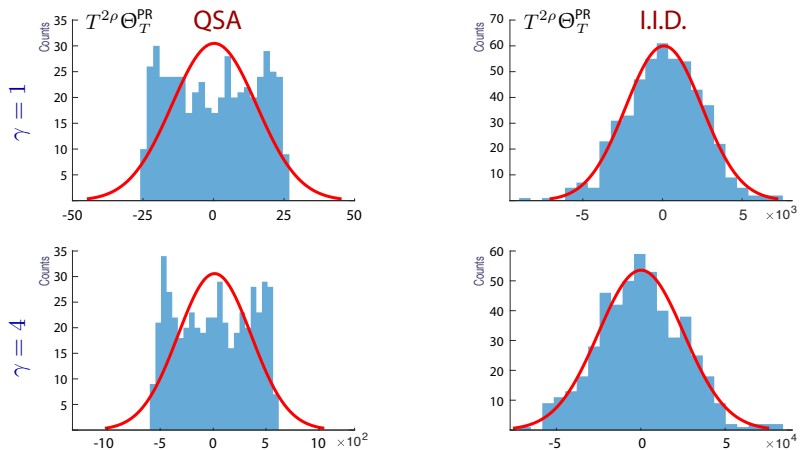

Figure 9: Comparison of Monte Carlo and quasi-Monte Carlo with Polyak-Ruppert averaging.

## C   Numerical Experiments

This final section is divided into two parts: a numerical experiment illustrating $O(T^{-2+\delta})$ convergence rate for QMC and details of the experiments surveyed in Section 3.2 of the main text.

### C.1   Quasi-Monte Carlo

Results from a simple experiment are provided here to illustrate that we can design algorithms to achieve convergence rates far faster than $O(\log(n)^K/n)$ in applications to QMC.

Suppose that $\xi$ is $m$-dimensional, with components equal to triangle waves: $\xi^i_t = \triangle(\omega_i t + \phi_i)$ for each $i$ and $t$, with $\triangle$ the unit sawtooth wave with unit period and range $\pm 1$:

$$\triangle(t) = 1 - 4|\tfrac{1}{2} - \mathrm{frac}\{t + \tfrac{1}{4}\}|$$

We present results from a simple experiment as illustration: Our goal is to estimate the mean of $g(Y)$ with $g\colon \mathbb{R}^2 \to \mathbb{R}$ and $Y$ uniformly distributed on the rectangle $[-1,1]^2$. The probing signal is chosen to be two-dimensional, so that a QSA algorithm to estimate $\overline{h}$ is obtained with $f(\theta, \xi) = -\theta + h(\xi)$ for which $\overline{f}(\theta) = -\theta + \overline{h}$:

$$\tfrac{d}{dt}\Theta_t = a_t[-\Theta_t + h(\xi_t)] \tag{63}$$

Based on the formula (20) we have $\overline{Y}^* = 0$.

Consider the function $h(x_1, x_2) = \exp(\gamma x_1)\sin(2\pi(x_2 - x_1))$, for which $\overline{h} = 0$ for any value $\gamma$. The frequencies $\{\omega_1, \omega_2\} = \{\log(6), \log(2)\} \approx \{1.8, 0.69\}$ were chosen, along with several values of $\rho$, and $\kappa = 5$ in application of PR averaging. The ODEs were approximated using an Euler approximation with sampling time $T_s = 0.1$ sec., which is roughly 1/5 of the shortest period $1/\omega_1$.

The data displayed in Fig. 9 is based on four experiments, differentiated by two values $\gamma = 1, 4$, and choice of probing signal. The first column uses the triangle wave described above, while the second is based on $\{\xi_{t_n} : n \ge 0\}$ i.i.d. and uniform on $[-1,1]^2$, where $\{t_n\}$ are the sampling times used in the Euler approximation for QSA. It is known that Polyak-Ruppert averaging shares the same CLT (asymptotic) variance as the usual sample path average when the probing signal is i.i.d. [35, 36]. The empirical variance using PR averaging was found to be similar to what was obtained using standard Monte Carlo.

Each histogram was created based on a runlength of $T = 10^4$ (corresponding to $10^5$ samples, given $T_s = 0.1$), and 500 independent runs in which the phases were sampled uniformly and independently on $[0, 1]$, and the initial condition $\theta_0$ was sampled uniformly and independently on $[-25, 25]$. The value $\rho = 0.7$ was chosen, resulting in $T^{2\rho} \approx 4 \times 10^5$. The MSE is roughly *four orders of magnitude larger* for the stochastic algorithm as compared to QSA.

The remaining experiments surveyed here are based on $\gamma = 1$.

Fig. 3 shows histograms of the scaled estimation error for quasi-Monte Carlo with averaging for two cases: $\rho = 0.7\,(\beta = 1.4)$ and $\rho = 0.8\,(\beta = 1.6)$. The scaled error exhibits more variability as $\rho$ is increased. In this experiment, the histogram obtained with $\rho = 0.8$ is roughly two times wider than for $\rho = 0.7$. In other words, the rate of convergence is bounded by a constant $B_\rho$ times $T^{-2\rho}$, and in these experiments we observe that the best constant $B_\rho$ is an increasing function of $\rho$.

Fig. 10 shows sample paths of estimates for standard Monte Carlo and the stochastic PR algorithm, with $\{\xi_{t_n} : n \geq 0\}$ i.i.d. and uniform.

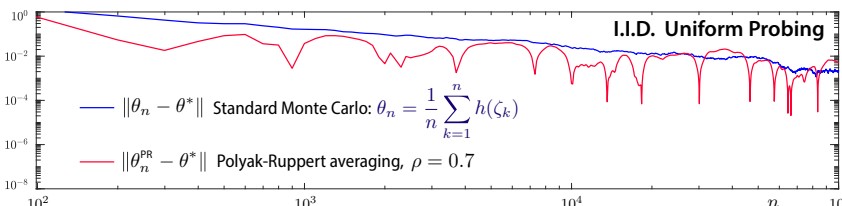

Figure 10: Slow convergence of Monte Carlo and Monte Carlo with Polyak-Ruppert qveraging.

The next set of experiments illustrate that the rate of convergence observed in Fig. 10 is *far slower* than observed in any of the deterministic algorithms.

Each row of Fig. 11 contains four plots: $|\Theta_T|$ obtained with (63) using the indicated value of $\rho$, $|\Theta_T^{\mathrm{PR}}|$ obtained using PR averaging, and $T^{-\rho}$, $T^{-2\rho}$ as these are the convergence rate bounds for each case. The plots illustrate that the $O(T^{-2\rho})$ bound on the convergence rate using PR averaging is achieved for smaller values of $\rho > 1/2$. With $\rho = 0.9$ the results are not so clearly compatible with theory; recall that the theory requires $\rho < 1$, so we expect numerical challenges when $\rho$ is close to 1.

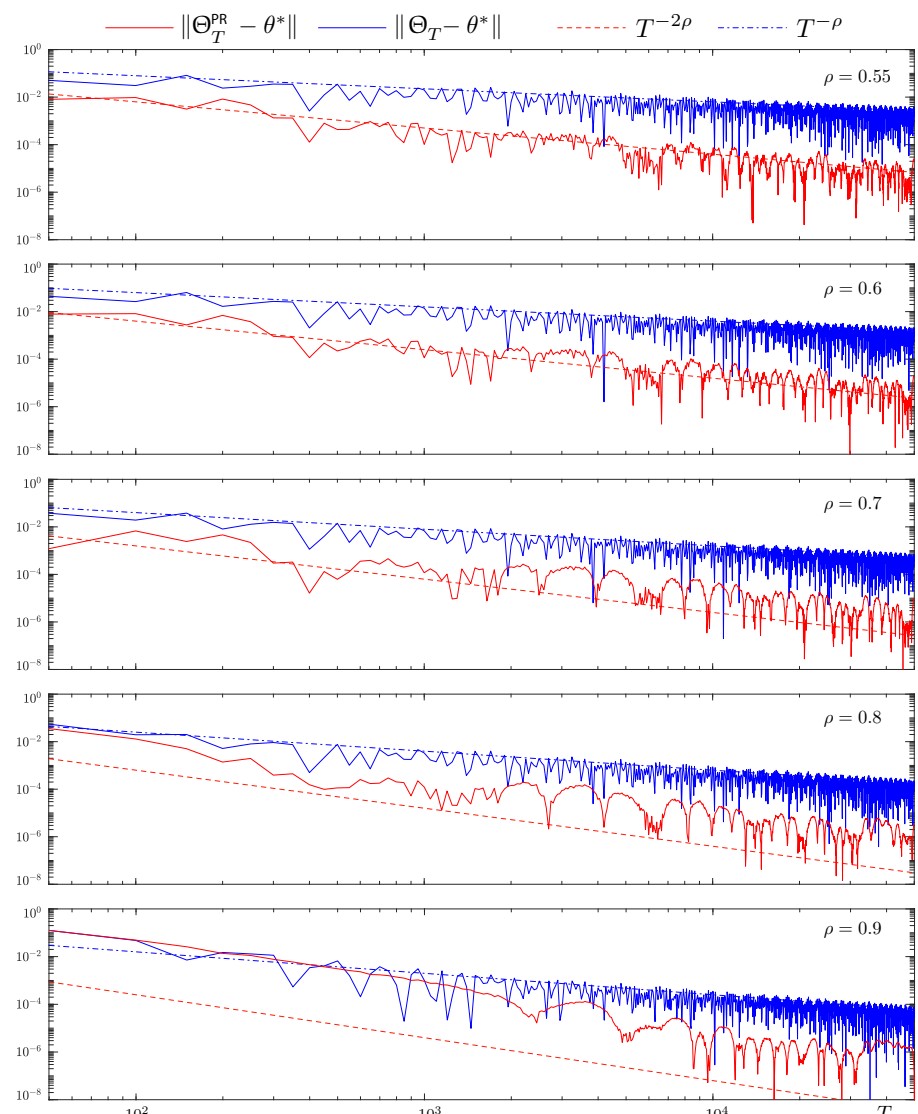

Figure 11: Quasi-Monte Carlo with Polyak-Ruppert averaging.

## C.2 Numerical experiments in Section 3.2

Here we provide full details regarding the numerical experiments in Section 3.2, whose objective functions were selected from [47].[1] For a rectangular region $B_0 \subseteq \mathbb{R}^d$, we will frequently use the expression "projection of sample-paths onto $B_0$". This means we project the trajectories $\{\Theta_t\}$ to $B_0$ component-wise in any approximation of the QSA ODE. For the one-dimensional setting with $B_0 = [-1, 1]$, the projection is defined by $\max\{-1, \min\{1, \theta_n\}\}$ where $\theta_n$ is an approximation of $\Theta_{t_n}$ at sampling time $t_n$.

Projection is often necessary because Lipschitz continuity of $f$ is assumed in (QSA2). This requires $\Gamma$ to be Lipschitz continuous when employing 1qSGD (23) and $\nabla\Gamma$ when using 2qSGD (24) [9, 30]. Restricting $\{\Theta_t\}$ to a closed and bounded set is a way to relax these requirements.

Each experiment contained the following common features:

(i) Only 1qSGD was considered.

(ii) The gain process was $a_t = \min\{a_0, (t+1)^{-\rho}\}$ with $1/2 \leq \rho \leq 1$, and $a_0 > 0$.

(iii) The QSA parameters $a_0$ and $\varepsilon$ were problem specific and chosen by trial and error.

(iv) The ODE (23) was approximated by an Euler scheme with sampling time equal to 1 sec. This crude approximation is justified via (ii) by choosing $a_0 > 0$ sufficiently small.

(v) Averaging was performed with $\kappa = 5$ (final 20% of samples).

(vi) A rectangular constraint region $B_0 \subseteq \mathbb{R}^d$ was fixed. The selection of $B_0$ was based on conventions of the particular objective given in [47].

(vii) For each objective and algorithm, the frequencies were held fixed in $M$ independent experiments: $\{\omega_i : 1 \leq i \leq d\}$ were uniformly sampled from $[0.05, 0.5]^d$. The initial conditions and phases were not held constant: For $\{1 \leq m \leq M\}$,

   (a) The phases $\{\phi_i^m : 1 \leq i \leq d\}$ were sampled uniformly on $[-\pi/2, \pi/2]^d$. That is, the probing signal respected (10), and in the $m$th experiment the probing signal took the form

$$\xi_t^m = 2[\sin(\omega_1 t + \phi_1^m), \cdots, \sin(\omega_d t + \phi_d^m)]^\mathsf{T}$$

   giving $\Sigma_\xi = 2I$.

   (b) The initial condition $\Theta_0^m$ were selected uniformly at random from $B_0$.

(viii) The outputs were the sample covariance (26), the sample paths $\{\Theta_t^m : t \geq 0\}$ and/or $\{\Gamma(\Theta_t^m) : t \geq 0\}$.

**Rastrigin** We return to a discussion regarding the experiments supporting Figs. 5 and 6. For this objective, $d = 2$, and experiments used $T = 8 \times 10^4$, $M = 200$ and $a_0 = 0.5$. The projection region was $B_0 = [-5.12, 5.12]^d$, following [47]. Finally, the noise for the stochastic algorithm was a scaled and shifted Bernoulli with $p = 1/2$ and support chosen so that its covariance matrix $\Sigma$ satisfied $\Sigma = \Sigma_\xi$.

We see in Fig. 5 that apart from achieving near quartic convergence rates, the estimates of $\theta^{\text{opt}}$ resulting from the deterministic algorithm had less variability than their stochastic counterpart. This reduction of variability was roughly two orders of magnitude. In both cases, the histograms show that the algorithms consistently overestimate $\theta^{\text{opt}}$. This is most likely due to the bias that is inherent in these algorithms (recall Section 3.1).

The plots in Fig. 12 show a comparison of both $\Gamma(\Theta_T)$ and $\Gamma(\Theta_T^{\text{PR}})$ for three distinct initial conditions $\Theta_0$. This plot illustrates the benefits of PR averaging as $\Gamma(\Theta_T^{\text{PR}})$ approaches $\Gamma(\theta^*) = 0$ much quicker than $\Gamma(\Theta_T)$ for each initial condition.

The experiment leading to the results in Fig. 6 used the same parameters described above but the run length chosen was $T = 4 \times 10^4$ for better visualization of the sample paths $\{\Theta_T\}$ and $\{\Theta_T^{\text{PR}}\}$.

---

[1]Publicly available code obtained under GNU General Public License v2.0.

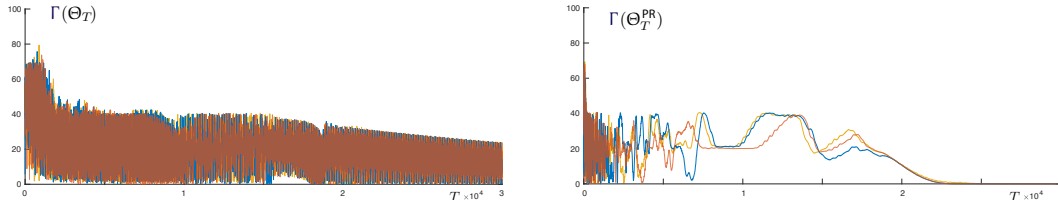

Figure 12: Evolution of $\Gamma(\Theta_T)$ (left) and $\Gamma(\Theta_T^{\mathsf{PR}})$ (right) as functions of $T$.

**Ackley** We performed $M = 50$ independent experiments of run length $T = 8 \times 10^4$ to minimize the Ackley objective with projection region equal to $B_0 = [-32.768, 32.768]^d$ [47]. Experiments were performed for two sets of parameters:

(i) $\rho = 0.85$, $a_0 = 0.02$ and $\varepsilon = 0.01$.

(ii) $\rho = 0.7$, $a_0 = 0.07$ and $\varepsilon = 0.1$.

For each set of parameters, the root mean square error $T^{2\rho}\sqrt{\text{tr}(\overline{\Sigma}_T)}$ was obtained for $d = 2$ and $d = 30$. Results for case (i) are pictured in Fig. 7 and repeated in Fig. 13 along with results for case (ii).

As mentioned in Section 3.2, the run length used was not sufficiently large for $\rho = 0.85$ with $d = 30$, but it was for $\rho = 0.7$. Simulations indicate that the variance is bounded by $O(a_T^4)$, but it is possible that their value grows with dimension. Better performance might be obtained by using different probing signals.

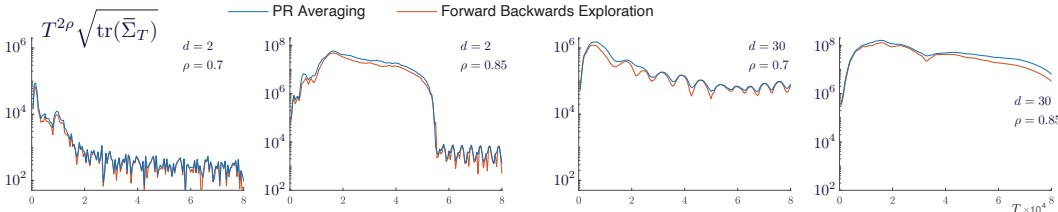

Figure 13: Comparison of scaled empirical covariances for Ackley objective.

**Three-Hump Camel** $M = 50$ independent experiments with run length $T = 8 \times 10^4$ were performed for this objective for $\rho = 0.85$ and $\rho = 0.7$. For both cases the QSA parameters used were $a_0 = 0.01$ and $\varepsilon = 0.5$. Here, $d = 2$ and $B_0 = [-5, 5]^d$ [47]. The average loss across initial conditions was obtained via

$$\bar{\Gamma}_T = \frac{1}{M} \sum_{m=1}^{M} \Gamma(\Theta_T^m)$$

The average loss and the root mean square error $T^{2\rho}\sqrt{\text{tr}(\overline{\Sigma}_T)}$ were obtained in each case and are shown by Fig. 8 and Fig. 14, respectively. The results for this experiment agree with the previous observations: near quartic rates are achieved. The variance appears to grow with $\rho$.

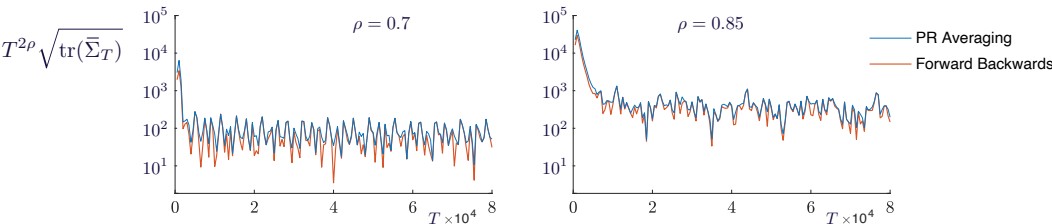

Figure 14: Scaled empirical covariances for $\rho = 0.7$ (left) and $\rho = 0.85$ (right) for three-hump camel objective.