# OpenReview forum: "Approaching Quartic Convergence Rates for Quasi-Stochastic Approximation with Application to Gradient-Free Optimization"
_NeurIPS.cc/2022/Conference — NeurIPS 2022 Accept_

### Official Review · Reviewer_JEVp · 2022-07-08

**Rating:** 5
**Confidence:** 4
**Soundness:** 3 good
**Presentation:** 2 fair
**Contribution:** 3 good

**Summary:**

In this paper, the authors focus on the theoretical analysis and the application of quasi-stochastic approximation to the gradient-free optimization. The authors utilized some acceleration techniques to achieve the faster rate of convergence of order $\mathcal{O}(n^{\delta - 4})$ where $\delta > 0$ is any arbitrary positive number. Then, the authors specialize this theoretical results to the setting of the gradient-free optimization problem. The authors also conduct several numerical experiments and these empirical results are consistent with the theoretical theorems. The authors applied the results from stochastic approximation theory and the most recent discovery from number theory to prove the theoretical theorems. All proof has been put in the supplementary material of the appendix.

**Questions:**

One question:

In line 18, is that convergence notation the notation of convergence in distribution? If it is, the authors should explicitly explain this notation and give an introduction to the concept of convergence in distribution, so that the readers could easily understand this.

**Limitations:**

The authors have presented the limitations of this paper in the final conclusion section.

**Strengths And Weaknesses:**

This paper has the following strengths in terms of the originality, quality, clarity and significance:

Originality and Significance: this paper provides the first analysis and proof of quadratic convergence rates of quasi-stochastic approximation with application in the gradient-free optimization. This is the main and independent contribution of this paper.

Quality: the authors provides detailed and correct theoretical analysis in the proof section and reasonable explanations in the numerical experiments section. These theoretical and empirical results are consistent with each other.

Clarity: the paper is organized with clear and clean structure and the language is well-organized so that the readers can easily understand the content of this submission.

From my perspective, there is no major weakness for this submission. One minor weakness is that the authors should put the literature review, notation and organization of section Introduction into another section so that the structure of the submission could be more clear.

---

### Official Review · Reviewer_dkZC · 2022-07-09

**Rating:** 7
**Confidence:** 3
**Soundness:** 3 good
**Presentation:** 3 good
**Contribution:** 3 good

**Summary:**

This paper deals with quasi-stochastic approximation -  i.e. when the random noise of standard stochastic approximation schemes is replaced by exponential signals - and provides novel and faster convergence rates. The authors introduced acceleration techniques to derive such theoretical results and highlight them in the context of gradient-free optimization. Numerical experiments support the theoretical findings.




**Questions:**



**Limitations:**

This is a theoretical work with no foreseeable negative societal impacts.

**Strengths And Weaknesses:**

First, I would like to thank the authors for their work: It is a well written and clear paper with relevant motivations (clear introduction), well-explained related work and fine overall structure. Although I am not an expert in this area of research, I found the message of the article to be clear and the theoretical results to be well presented and discussed.

- Originality: The idea of forward-backward filtering is originial and there are solid contributions. This paper presents a new framework for quasi-stochastic approximation which extends several convergence results known so far in restricted settings. The related work is adequately cited and clearly describes how the current paper differs from it.

- Quality: Technically strong, highly general results, advanced techniques.

- Clarity: The paper is very well written with fine overall structure making it easy for the reader to grasp all the main ideas. The authors may consider to add and discuss the following works which are strongly related to the current contribution:

     * \[A] "Optimal rate of convergence for quasi-stochastic approximation". (Bernstein, Andrey and Chen, Yue and Colombino, Marcello and Dall'Anese, Emiliano and Mehta, Prashant and Meyn, Sean)
     * \[B] "Accelerating optimization and reinforcement learning with quasi stochastic approximation". (Chen, Shuhang and Devraj, Adithya and Bernstein, Andrey and Meyn, Sean)

- Significance: The results are quite important as they extend and relax several assumptions made so far in the quasi-stochastic approximation litterature.

**References**

- \[A]: @article{bernstein2019optimal,
  title={Optimal rate of convergence for quasi-stochastic approximation},
  author={Bernstein, Andrey and Chen, Yue and Colombino, Marcello and Dall'Anese, Emiliano and Mehta, Prashant and Meyn, Sean},
  journal={arXiv preprint arXiv:1903.07228},
  year={2019}}

- \[B]: @inproceedings{chen2021accelerating,
  title={Accelerating optimization and reinforcement learning with quasi stochastic approximation},
  author={Chen, Shuhang and Devraj, Adithya and Bernstein, Andrey and Meyn, Sean},
  booktitle={2021 American Control Conference (ACC)},
  pages={1965--1972},
  year={2021},
  organization={IEEE}}

---

### Official Review · Reviewer_DYiM · 2022-07-12

**Rating:** 7
**Confidence:** 4
**Soundness:** 3 good
**Presentation:** 3 good
**Contribution:** 3 good

**Summary:**

This manuscript provides a fast convergence rate analysis for nonlinear quasi-stochastic approximation by the Polyak-Ruppert averaging technique. The main ingredient for achieving acceleration is to carefully design the sine type probing signals to break the $O(n^{-2})$ bound. The application of this idea to gradient-free optimization is also discussed. Some experiments are conducted on simple test functions and it seems to verify the theory.

**Questions:**

1. Is it possible to relax the requirement of the probing signal (e.g., to a broader class of signals) and still achieve acceleration?

2. How to translate the Quasi-Stochastic Approximation ODE to a stochastic algorithm?



**Strengths And Weaknesses:**

Strengths:
(i) The paper is well-written. The main message of the paper is very clear.
(ii) The theoretical result looks sound and strong. The overview of the proof makes sense to me (although I did not read all the proofs in the appendix in great detail).

Weaknesses:
(i) The specific form of the probing signal is very important to achieve acceleration. However, the noise design is not realistic in machine learning applications. For example, in standard stochastic approximation or SGD for training machine learning models, we do not really have the knowledge of the noise. Some discussions/clarifications would be better.

(ii) The whole paper considered an ODE formulation, it is unclear whether there is a corresponding stochastic algorithm that has a fast convergence rate as well.

---

### Official Review · Reviewer_SkSg · 2022-07-13

**Rating:** 7
**Confidence:** 2
**Soundness:** 3 good
**Presentation:** 4 excellent
**Contribution:** 3 good

**Summary:**

This paper proposes to solve the root finding problem of the form $\bar f(\theta^* ) = 0$ where $\bar f$ is the expectation of some smooth but random function. Instead of the usual ODE method that converges at at rate of $O(1/t)$ in the MSE, the authors propose to use QSA ODE instead to obtain a convergence rate of $O(1/t^2)$, and *close to* $O(1/t^4)$ if combined with Polyak-Rupert (PR) Averaging. The QSA ODE method is similar to the ODE method but with sinusoidal perturbations rather than iid uniform perturbations. The theoretical results provide insights to new approaches in Quasi Monte Carlo sampling. Finally, simulation studies using Euler approximation corroborate with the theoretical results.

**Questions:**

Please see section above.

**Limitations:**

I find the discussions on limitations to be fairly adequate. This is particularly reflected in Section 2.1, where the authors discuss bounds on the constant $K_f$.


**Strengths And Weaknesses:**

I find this submission to be very interesting, unfortunately, I would like to first acknowledge that after carefully reading the paper, I realized my expertise in certain technical aspects is lower than I had anticipated, especially when it comes to complex analysis, the invocation of Baker's Theorem, and assumptions on the probing signal. I only looked through the proofs at a high level and was unable to understand every detail. My evaluation is thus also at a very high level based on my knowledge in other aspects such as discrete time stochastic optimization and QMC. I look forward to learning more from the authors and the other reviewers.

I particularly enjoy the new QMC results developed that give rise to Theorem 2.1 (continuous time) and B.9 (discrete time). One concern I have is that the authors claim (line 87-89) the latter result gives a $O(1/n)$ instead of the usual $O(log(n)^d/n)$ rate for QMC in the root MSE. To me, removing a dependence on the dimensionality $d$ is a huge improvement, as QMC is long known to work better than MC only in the low dimensional setting. However, as the authors pointed out, the $O(1/n)$   actually contains a multiplicative constant $K_f$ for which known bounds grow at least exponentially in $d$. It is thus unclear to me whether this is in fact a better rate than classical QMC rates. If yes, it would be great if the authors could elaborate more on precisely what settings this would be the case.

Although PR Averaging has long been studied in stochastic optimization, I have not seen it being used in conjunction with Quasi-Monte Carlo perturbations. The near-quartic convergence obtained from this is therefore impressive, and it would be interesting to see if similar techniques (such as using sinusoidal noise) can be incorporated to, say, SGD, to obtain even faster rates.

Overall, I think the submission is well polished and organized. The experiments setups are justified and plots are nicely presented, with details clearly stated in the appendix.

---

### Author Response · Authors · 2022-08-02
**Author Response for Paper3691**

We appreciate the feedback from all of the reviewers and will take all of their input into account in a revision.
Since the submission we realized that more theory should be on display in the body of the paper. In particular, the following exact representation of the QSA ODE
$$
		\frac{d}{dt} \Theta_t = a_t \Big[ \bar f (\Theta_t) - a_t \bar\Upsilon_t + a_t^2 W_t^0 + a_t \frac{d}{dt} W^1 _t + \frac{d^2}{dt^2} W^2_t \Big]
		\qquad {(\star)}
$$
in which {$\bar\Upsilon_t , W^i _t : i=0,1,2 $} are smooth functions of $(t, \Theta_t)$. The proof of Theorem 2.2 follows, as does Prop. 2.3 (derivatives are attenuated upon averaging). This representation provides other insights regarding transient behavior, and will help us respond to comments from the reviewers.

We were pleased to read the reviewers' appreciation of both exposition and content (in particular, the near quartic convergence rate). The analysis is also novel: Baker's theorem has been with us for 50 years without application to learning; we can find no other approach to obtain the sharp bounds for rates of convergence. The multiple applications of Poisson's equation to obtain ($\star$) is entirely novel.

$\textbf{Responses to rev. SkSg}$:

$\textit{Impact of dimension.}$ Agreed: the QMC literature obtains a rate of convergence of order $O(\log(n)^d n^{-1})$. We dramatically beat this in terms of $n$-dependency [rate of order $O(n^{-2+\delta})$], but we do not have theory to predict the impact of dimension. In a revision we will emphasize this gap in current theory. However, comparison with traditional QMC is not straightforward. To the best of our knowledge, the QMC community has not addressed nonlinear dynamical systems similar to QSA. Moreover, if we move to exploration based on traditional QMC then we lose the crucial geometry used to obtain our main results.

$\textit{Application to SGD.}$ There are many dimensions to this question. For SGD with truly noisy observations, we must face the Cramer-Rao lower bound (of Polyak and Ruppert), which means that the MSE converges at rate $n^{-1}$ rather than $n^{-4+\delta}$. In other formulations of SGD such as block gradient descent, the covariance of the noise is state dependent and vanishes at the stationary point. In such cases we may see significant benefit from representations similar to ($\star$). See replies to rev. DYiM regarding stochastic approximation for further thoughts.

$\textbf{Responses to rev. DYiM:}$

$\textit{Knowledge of the noise.}$ Absolutely---we agree entirely that application of QSA is not universal. We argue that there is a perfect fit with gradient free optimization, and more limited instances of reinforcement learning (when "noise" is largely designed by the user for exploration). The efficiency and reliability in applications to optimization is remarkable in all of our experiments so far, especially when compared with the standards such as SPSA (cf. figure 4).

$\textit{Implication to stochastic recursions.}$ We also had thoughts on this point, and since the submission we have constructed a representation similar to ($\star$) for stochastic recursions in discrete time. We find that the nuisance term $\bar\Upsilon$ appears when there is multiplicative noise (such as in TD-learning). It has strongly negative implications, especially for fixed gain algorithms. This is beyond the scope of this submission, but we will add remarks.
Beyond this, we wonder if there is any hope in finding stochastic probing signals that exhibit similar geometry to what is found in QSA. The orthogonality of functions of the probing signal and solutions to Poisson's equation was crucial to eliminate $\bar\Upsilon$. We are currently considering classes of reversible Markov chains, but have not yet had success.

Please see our response to rev. SkSg for more on this topic. In particular, due to the central limit theorem we can only expect $n^{-1}$ rate of convergence for the MSE, rather than $n^{-4+\delta}$ obtained here.

$\textit{Broader classes of probing signals.}$ We agree that we must go beyond mixtures of sinusoids in applications to gradient free optimization. The general theory in this paper allows for any smooth function of sinusoids with irrationally related frequencies, so we can approximate triangle or sawtooth waves, or more chaotic signals (though we must respect physics in online optimization). We do not yet have theory to predict an optimal choice for probing.

$\textbf{Responses to rev. dkZC:}$ We will update the references to include those recommended, and also past ML articles on zeroth order optimization (mainly to help put this work in context with articles familiar to the NeurIPS audience).

$\textbf{Responses to rev. JEVp:}$ We will create a section dedicated to literature review, notation and organization. In particular, the notation on line 18 does refers to convergence in distribution, so a statement will be added to avoid confusion.

---

> ### Comment · Reviewer_SkSg · 2022-08-07
> **Comment on Author Response**
>
> I appreciate the response and agreed that traditional QMC may not yield the accelerated rate obtained in this paper. It would still be very helpful to emphasize and elaborate on the hidden dependence on $d$ in the rate. I am leaning towards increasing my rating, however, I would like to acknowledge again that I did not understand a majority of the proofs.

---

> > ### Author Response · Authors · 2022-08-08
> > **Response to SkSg**
> >
> > You may be reassured to learn that we discovered an error in our statement on dependencies: the exponential bound in Baker’s Theorem concerns $K$ (the number of frequencies) and NOT $d$. Origin of error: we took $K=d$ to simplify notation. We apologize for the confusion this caused!
> >
> > To avoid assuming $K=d$ we will make a change in notation: First, denote by $\xi_t^0$ the vector of $K$ sinusoids of differing frequencies, and then $\xi_t = G_0(\xi_t^0 )$ with $G_0$ analytic. Example: $K=1$ and $G_0$ polynomial to obtain
> > $\xi_t = [\cos(\omega t), \cos(2\omega t), \dots, \cos(d\omega t) ]$.
> >
> > To the best of our knowledge, and experts we have consulted, the QMC community does not consider bounds for QSA, only Monte-Carlo. We will make clear in the revision that dimension dependent bounds for QSA is an entirely open field for research.
> >
> > The paper is a challenge to review because of the unusual mix of mathematical techniques.  The disturbance decomposition begins with an old trick from the SA literature, but the multiple applications of Poisson's equation is entirely novel (and essential).  The geometry surrounding Poisson's equation is entirely new.  And, as remarked in the rebuttal letter, we have not seen application of Baker's theorem in prior research in any ML related domain.  To help the reader we plan to move essential elements of the theory to the body, including:
> > 1. The representation of the QSA ODE (see rebuttal).
> > 2. The geometry surrounding Poisson’s equation from the supplementary material (currently hidden in Corollary B.5).
> > 3. Clarity regarding the role of Baker's Theorem to obtain bounds  (as explained above Theorem B.6, this is used to obtain a lower bound on $| \omega_{\alpha,b} |$ that is essential to bound solutions to Poisson's equation).
> >
> > We can make space by moving discussion on Monte-Carlo to the supplementary material.

---

### Meta-Review · Area_Chair_fHVu · 2022-08-26

**Recommendation:** Accept
**Confidence:** Certain

**Metareview:**

All reviewers are happy with the new ideas and strength of results in this paper.

**Award:**

No

---

### Decision · Program_Chairs · 2022-09-14

Accept